# Video Diffusion Models: A Survey

**Andrew Melnik**                                    *andrew.melnik.papers@gmail.com*
*Bielefeld University*

**Michal Ljubljanac**                          *mljubljanac@techfak.uni-bielefeld.de*
*Bielefeld University*

**Cong Lu**                                                    *conglu@cs.ubc.ca*
*University of British Columbia*

**Qi Yan**                                                      *qi.yan@ece.ubc.ca*
*University of British Columbia*

**Weiming Ren**                                              *w2ren@uwaterloo.ca*
*University of Waterloo*

**Helge Ritter**                                    *helge@techfak.uni-bielefeld.de*
*Bielefeld University*

**Reviewed on OpenReview:** *https://openreview.net/forum?id=rJSHjhEYJx*

## Abstract

Diffusion generative models have recently become a powerful technique for creating and modifying high-quality, coherent video content. This survey provides a comprehensive overview of the critical components of diffusion models for video generation, including their applications, architectural design, and temporal dynamics modeling. The paper begins by discussing the core principles and mathematical formulations, then explores various architectural choices and methods for maintaining temporal consistency. A taxonomy of applications is presented, categorizing models based on input modalities such as text prompts, images, videos, and audio signals. Advancements in text-to-video generation are discussed to illustrate the state-of-the-art capabilities and limitations of current approaches. Additionally, the survey summarizes recent developments in training and evaluation practices, including the use of diverse video and image datasets and the adoption of various evaluation metrics to assess model performance. The survey concludes with an examination of ongoing challenges, such as generating longer videos and managing computational costs, and offers insights into potential future directions for the field. By consolidating the latest research and developments, this survey aims to serve as a valuable resource for researchers and practitioners working with video diffusion models. Website: https://github.com/ndrwmlnk/Awesome-Video-Diffusion-Models

## 1 Introduction

Diffusion generative models (Sohl-Dickstein et al., 2015; Song & Ermon, 2019; Ho et al., 2020; Song et al., 2021; Ruiz et al., 2024) have already demonstrated a remarkable ability for learning heterogeneous visual concepts and creating high-quality images conditioned on text descriptions (Rombach et al., 2022; Ramesh et al., 2022). Recent developments have also extended diffusion models to video (Ho et al., 2022c), with the potential to revolutionize the generation of content for entertainment or simulating the world for intelligent decision-making (Yang et al., 2023a). For example, the text-to-video SORA (Brooks et al., 2024) model has been able to generate high-quality videos up to a minute long conditional on a user's prompt. Following

the announcement of SORA, there has been a surge of proprietary and open-source video diffusion models spanning across a wide variety of video generation problems, such as text-to-video generation, image-to-video generation, and video editing. Several commercial AI video generation tools are gaining attention for their unique features. Runway's Gen-3 stands out with highly photorealistic videos, especially for complex scenes and faces, offering both text-to-video and image-to-video capabilities with professional-level quality. Luma Dream Machine, though less coherent in complex scenes, remains user-friendly and affordable for quick cinematic outputs. Kling AI, with its mobile-first approach, provides strong control over elements like lighting and motion, while ensuring high video generation quality. Movie Gen (Polyak et al., 2024) from Meta offers a suite of foundation models capable of generating high-quality, high-resolution videos in various aspect ratios with synchronized audio. For open-source models, recent foundational video generation models include OpenSora (Zheng et al., 2024), VideoCrafter-2 (Chen et al., 2024) and Stable Video Diffusion (Blattmann et al., 2023a). However, the quality, resolution and duration of the generated videos are still not comparable to commercial video generation solutions. Adapting diffusion models to video generation poses unique challenges that still need to be fully overcome, including maintaining temporal consistency, generating long video, and computational costs.

In this survey, we provide an overview over key aspects of video diffusion models including possible applications, the choice of architecture, mechanisms for modeling of temporal dynamics, and training modes (see Figure 1 and Table 1 for an overview). We then provide brief summaries of notable papers in order to outline developments in the field until now. Finally, we conclude with a discussion of ongoing challenges and identify potential areas for future improvements.

## 2    Taxonomy of Applications

The possible applications of video diffusion models can be roughly categorized according to input modalities. This includes text prompts, images, videos, and auditory signals. Many models also accept inputs that are a combination of some of these modalities. Figure 2 visualizes the different applications. We summarize notable papers in each application domain starting from Sec. 7.1.3. For this, we have categorized each model according to one main task.

In our taxonomy, *text-conditioned generation* (Sec. 7.1.3) refers to the task of generating videos purely based on text descriptions. Different models show varying degrees of success in how well they can model object-specific motion. We thus categorize models into two types: those capable of producing simple movements such as a slight camera pan or flowing hair, and those that can represent more intricate motion over time, such as those incorporating Physical Reasoning (Melnik et al., 2023).

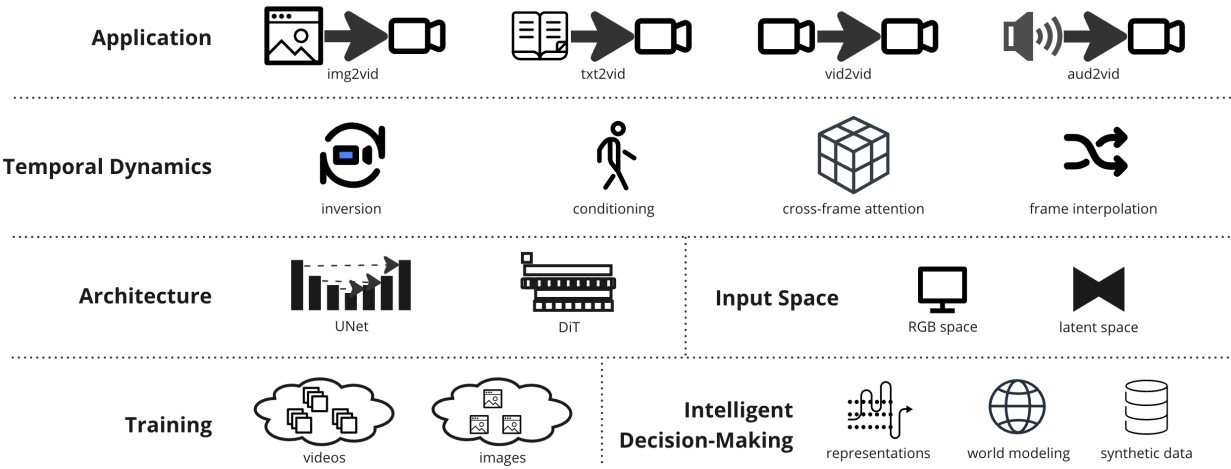

Figure 1: Overview of the key aspects of video diffusion models that we cover in this survey.

Table 1: Overview of video diffusion models and their applications.

| Paper | Model | Application | Max. Resolution | Methodology | Shots |
|---|---|---|---|---|---|
| Ho et al. (2022c) | VDM | T V L | 128×128×64 | FA ↑S ↑T AR | Many |
| Singer et al. (2022) | Make-a-Video | T I V | 768×768×76 | FA ↑S ↑T | Many |
| Ho et al. (2022a) | ImagenVideo | T | 1280×768×128 | FA ↑S ↑T | Many |
| Zhou et al. (2022) | MagicVideo | T I V | 1024×1024×61 | P L FA ↑S ↑T | Many |
| Blattmann et al. (2023b) | VideoLDM | T V | 2048×1280×90000 | P L FA ↑S ↑T AR | Many |
| Khachatryan et al. (2023) | Text2Video-Zero | T V | 512×512×8+ | P L | Many |
| Guo et al. (2023) | AnimateDiff | T | 512×512×16 | P L FA | Many |
| Chen et al. (2023d) | MCDiff | I | 256×256×10 | L AR | Many |
| Chen et al. (2023e) | SEINE | I | 512×320×16 | P L AR | Many |
| Yin et al. (2023) | Nuwa-XL | T L | NaN×NaN×1024 | P L FA ↑T | Many |
| He et al. (2022b) | LVDM | T L | 256×256×1024 | P L FA ↑T AR | Many |
| Harvey et al. (2022) | FDM | V L | 128×128×15000 | P FA ↑T AR | Many |
| Lu et al. (2023b) | VDT | I V | 256×256×30 | L FA ↑T | Many |
| Wang et al. (2023) | Gen-L-Video | T V L | 512×512×hundreds | P L 3D | NaN |
| Zhu et al. (2023) | MovieFactory | T L | 3072×1280×NaN | P L FA ↑S | Many |
| Sun et al. (2023) | GLOBER | T L | 256×256×128 | P L 3D FA | Many |
| Luo et al. (2023) | VideoFusion | T L | 128×128×512 | P ↑S AR | Many |
| Hu et al. (2023a) | GAIA-1 | T I L | 288×512×minutes | FA ↑T AR | Many |
| Lee et al. (2023a) | Soundini | A | 256×256×NaN | P | One |
| Lee et al. (2023b) | AADiff | I A | 512×512×150 | P L | Zero |
| Liu et al. (2023d) | Generative Disco | A | 512×512×NaN | P L | Zero |
| Tang et al. (2023) | Composable Diffusion | T I V A | 512×512×16 | P L FA | Many |
| Stypułkowski et al. (2023) | Diffused Heads | A | 128×128×8-9s | AR | Many |
| Zhua et al. (2023) | (Audio Heads) | A | 1024×1024×NaN | P ↑S AR | Many |
| Casademunt et al. (2023) | Laughing Matters | A | 128×128×50 | FA AR | Many |
| Molad et al. (2023) | Dreamix | I V | 1280×768×128 | P FA ↑S ↑T | One |
| Wu et al. (2022b) | Tune-A-Video | V | 512×512×100 | P L FA AR | One |
| Qi et al. (2023) | FateZero | V | 512×512×100 | P L FA AR | One |
| Liu et al. (2023b) | Video-P2P | V | 512×512×100 | P L FA AR | One |
| Ceylan et al. (2023) | Pix2Video | V | 512×512×NaN | P L FA AR | Zero |
| Esser et al. (2023) | Runway Gen-2 | T I V | 448×256×8 | P L FA | Many |
| Xing et al. (2023a) | Make-Your-Video | V | 256×256×64 | P L FA AR | Many |
| Ma et al. (2023) | Follow Your Pose | V | 512×512×100 | P L FA AR | Many |
| Zhao et al. (2023) | Make-A-Protagonist | V | 768×768×8 | P L FA | One |
| Bai et al. (2024) | UniEdit | V | 512×320×16 | P L | Zero |
| Ku et al. (2024) | AnyV2V | V | 512×512×16 | P L | Zero |
| Zhang et al. (2023c) | ControlVideo | V | 512×512×100 | P L 3D ↑T | Zero |
| Wang et al. (2023b) | vid2vid-zero | V | 512×512×8 | P L AR | Zero |
| Huang et al. (2023) | Style-A-Video | V | 512×256×NaN | P L | Zero |
| Yang et al. (2023b) | Rerender A Video | V | 512×512×NaN | P L ↑T AR | Zero |
| Liu et al. (2023a) | ColorDiffuser | V | 256×256×NaN | P L FA | Many |

T: txt2vid, I: img2vid, V: vid2vid, A: aud2vid, L: long vid

P: pre-trained model, L: latent space, 3D: full 3D attn./conv., FA: factorized attn./conv.,

↑S: spatial upsampling, ↑T: temporal upsampling, AR: auto-regressive

In *image-conditioned video generation* (Sec. 7.2) tasks, an existing reference image is animated. Sometimes, a text prompt or other guidance information is provided. Image-conditioned video generation has been extensively studied recently, due to its high controllability to the generated video content. For models introduced in other sections, we mention their capability for *image-to-video* generation where applicable.

We treat *video completion* (Sec. 8) models that take an existing video and extend it in the temporal domain as a distinct group, even though they intersect with the previous applications. Video diffusion models typically have a fixed number of input and output frames due to architectural and hardware limitations. To extend such models to generate videos of arbitrary length, both auto-regressive and hierarchical approaches have been explored.

*Audio-conditioned* models (Sec. 9) accept sound clips as input, sometimes in combination with other modalities such as text or images. They can then synthesize videos that are congruent with the sound source. Typical applications include the generation of talking faces, music videos, as well as more general scenes.

*Video editing* models (Sec. 10) use an existing video as a baseline from which a new video is generated. Typical tasks include style editing (changing the look of the video while maintaining the identity of objects), object / background replacement, deep fakes (Sec. 12), and restoration of old video footage (including tasks such as denoising, colorization, or extension of the aspect ratio).

Finally, we consider the application of video diffusion models to *intelligent decision-making* (Sec. 11). Video diffusion models can be used as simulators of the real world, conditioned on the current state of an agent or a high-level text description of the task. This could enable planning in a simulated world, as well as fully training reinforcement learning policies within a generative world model.

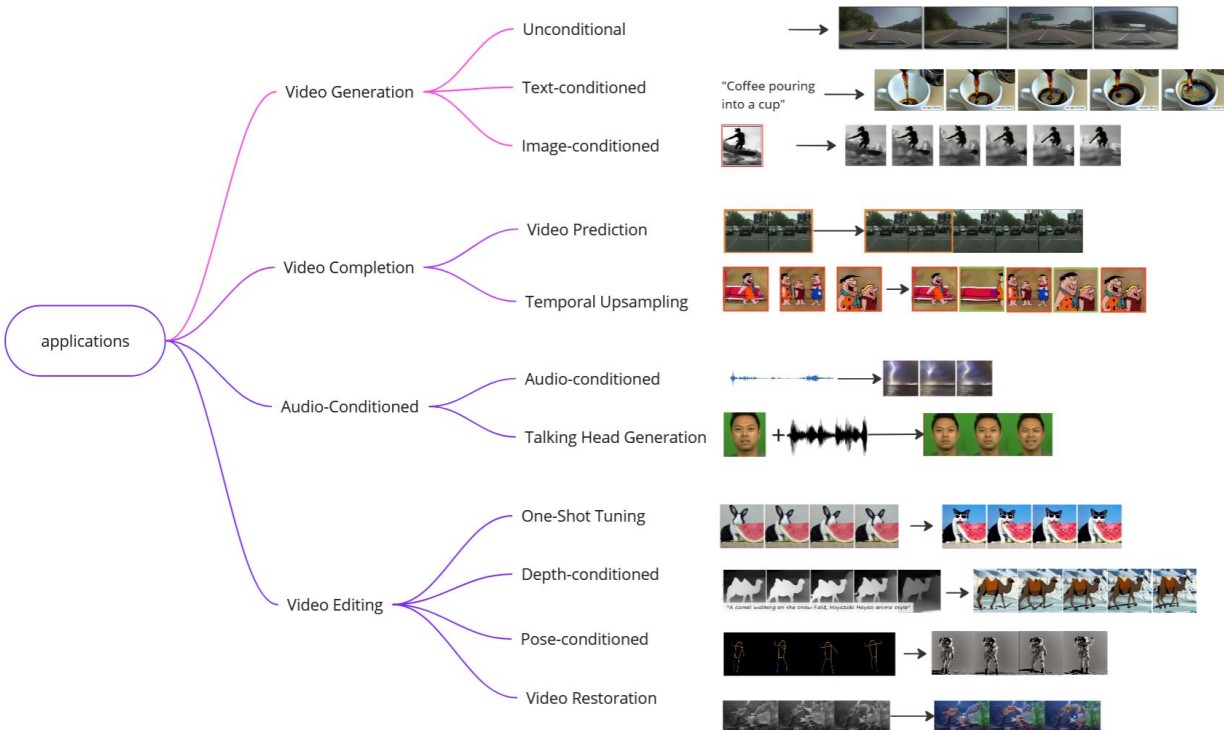

Figure 2: Applications of video diffusion models. Bounding boxes are clickable links to relevant chapters. Example images taken from the following papers (top to bottom): Blattmann et al. (2023b), Ho et al. (2022a), Singer et al. (2022), Lu et al. (2023b), Yin et al. (2023), Lee et al. (2023b), Stypułkowski et al. (2023), Wu et al. (2022b), Xing et al. (2023a), Ma et al. (2023), Liu et al. (2023a)

We conduct a thorough search for relevant papers on video diffusion models. This search includes open lists of accepted papers from recent conferences and journals in computer vision and machine learning, such as CVPR and NeurIPS. We select papers based on their relevance to the topic and their publication status, including archived and in-proceeding papers. Additionally, we utilize specific search terms related to video diffusion models in our database queries. Our search spans multiple reference databases, including Google Scholar and Semantic Scholar, to ensure we capture the latest preprint papers and ongoing research in this rapidly evolving field.

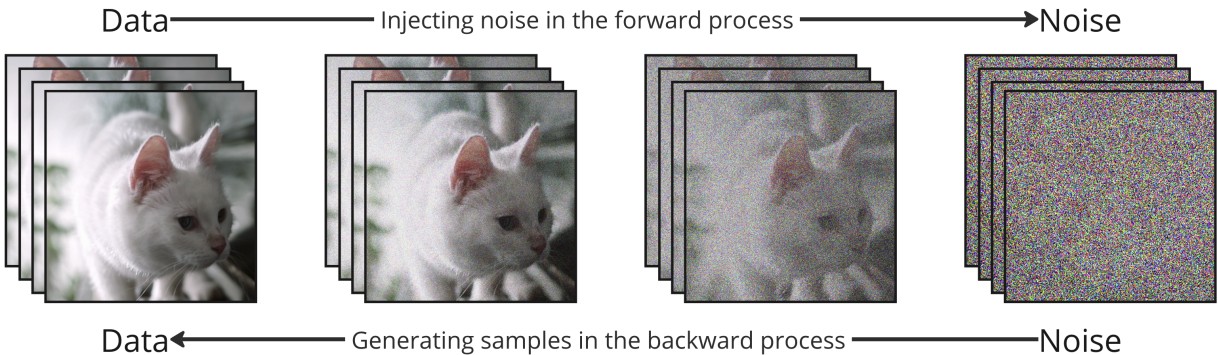

Figure 3: Diffusion models add noise to the observed data in the forward process and are trained to learn how to reverse the process. Denoising diffusion probabilistic model (DDPM) and score-based model (SBM) are popular approaches that are mathematically equivalent but provide different perspectives.

## 3 Mathematical Formulation

We first review the mathematical formulation of diffusion generative models, which learn to model a target distribution, for example, of natural videos, by progressive noise injection in the forward process and reversed denoising in the backward process, as shown in Fig. 3. A diffusion model generates samples via a chain of denoising steps that start from an initial noise vector that is often a sample from a Gaussian distribution of uncorrelated white noise. Each denoising step is performed by a neural network that has been trained to guide a noisy input toward the target distribution. After a number of such denoising steps, the obtained result will approximate a noise-free sample of the target domain. The key to this mechanism's success is training a suitable denoising network. This is achieved by an objective that learns to reverse the forward noising process at the corresponding noise levels. Broadly speaking, diffusion models can be categorized into two families of formulations: denoising diffusion probabilistic models (DDPM) (Ho et al., 2020; Sohl-Dickstein et al., 2015) and score-based model (SBM), the latter of which includes the denoising score matching models with Langevin dynamics (SMLD) (Song & Ermon, 2019; 2020) and the generalized Score SDE (Song et al., 2021). We give a brief introduction to each of the modeling designs below.

### 3.1 Denoising Diffusion Probabilistic Model (DDPM) Formulation

In the following, we summarize the formalization of the unconditioned denoising diffusion probabilistic model (DDPM) process from Ho et al. (2020); Sohl-Dickstein et al. (2015). The forward diffusion process follows a Markov chain that iteratively adds sampled noise to an initial input video $x_0$ over $T$ time steps, so that it resembles samples drawn from a Gaussian distribution. The Markov property ensures that the degraded video $x_t$ at time step $t$ only depends on the video $x_{t-1}$ in the immediately preceding step $t-1$. The distribution $q(x_t|x_{t-1})$ of $x_t$ in a forward step can be described by the conditional Gaussian transition kernel

$$q(x_t|x_{t-1}) := \mathcal{N}(x_t; \sqrt{1-\beta_t}x_{t-1}, \beta_t\mathbf{I}), \tag{1}$$

where the mean and standard deviation are determined by a variance-preserving noise schedule $\beta_1, ..., \beta_T$ and $\mathbf{I}$ is the identity matrix. Different schedules can be used, such as a linear or cosine schedule, influencing

how quickly information in the original vide is destroyed. Due to the Markov property that each state only depends on the preceding state, the overall forward process is described by

$$q(x_{1:T}|x_0) := \prod_{t=1}^{T} q(x_t|x_{t-1}). \tag{2}$$

Ho et al. (2020) show that the distribution at an arbitrary time step $t$ can be directly computed by

$$q(x_t|x_0) = \mathcal{N}(x_t; \sqrt{\overline{\alpha}_t}x_0, (1 - \overline{\alpha}_t)\mathbf{I}), \tag{3}$$

where $\overline{\alpha}_t := \prod_{s=1}^{t} \alpha_s$ and $\alpha_t := (1 - \beta_t)$. The analytical solution in the forward process allows for fast sampling of corrupted samples and reduces training variance.

In the denoising phase, we try to reverse this process, starting from the final time step $T$. The reverse process is again a Markov chain, this time with Gaussian transition probabilities that need to be learned by our model. A single denoising step is described by

$$p_\theta(x_{t-1}|x_t) := \mathcal{N}(x_{t-1}; \mu_\theta(x_t, t), \Sigma_\theta(x_t, t)), \tag{4}$$

where $\theta$ are the trainable parameters of our denoising model used to predict the mean and covariance of the conditional Gaussian. The full reverse process is described by

$$p_\theta(x_{0:T}) := p(x_T) \prod_{t=1}^{T} p_\theta(x_{t-1}|x_t), \tag{5}$$

where $p(x_T) := \mathcal{N}(x_T; \mathbf{0}, \mathbf{I})$ is the prior distribution to start drawing samples from in the reverse process.

To train the model, we minimize the variational lower bound on the negative log-likelihood

$$\mathbb{E}[-\log p_\theta(x_0)] \leq \mathbb{E}_q \left[ -\log p(x_T) - \sum_{t>1} \log \frac{p_\theta(x_{t-1}|x_t)}{q(x_t|x_{t-1})} \right]. \tag{6}$$

This loss function can be rewritten as a sum of Kulback-Leibler divergences between the distributions of the forward and backward steps

$$\mathcal{L} := \mathbb{E}_q \left[ D_{KL}(q(x_T|x_0)\|p(x_T)) + \sum_{t>1} D_{KL}(q(x_{t-1}|x_t, x_0)\|p_\theta(x_{t-1}|x_t)) - \log(p_\theta(x_0|x_1)) \right]. \tag{7}$$

This formulation has the advantage that we can calculate closed-form solutions for the Kulback-Leibler terms. First, the conditional distribution $q(x_T|x_0)$ admits an analytical solution in Eq. 3, and the reverse process prior $p(x_T)$ is fixed, making the first term $D_{KL}(q(x_T|x_0)\|p(x_T))$ a constant. One can design the noise schedule properly so that the terminal state in the forward process $q(x_T|x_0)$ is empirically close to the prior in the reverse process $p(x_T)$ to lower this term.

Note that the forward posteriors are now also conditioned on the initial video $x_0$. Using Bayes' theorem, it can be shown that

$$q(x_{t-1}|x_t, x_0) = \frac{q(x_t|x_{t-1}, x_0)\, q(x_{t-1}|x_0)}{q(x_t|x_0)} = \frac{q(x_t|x_{t-1})\, q(x_{t-1}|x_0)}{q(x_t|x_0)} = \mathcal{N}(x_{t-1}; \tilde{\mu}_t(x_t|x_0), \tilde{\beta}_t\mathbf{I}), \tag{8}$$

where $\tilde{\mu}(x_t, x_0) := \frac{\sqrt{\overline{\alpha}_{t-1}}\beta_t}{1-\overline{\alpha}_t}x_0 + \frac{\sqrt{\alpha_t}(1-\overline{\alpha}_{t-1})}{1-\overline{\alpha}_t}x_t$ and $\tilde{\beta}_t := \frac{1-\overline{\alpha}_{t-1}}{1-\overline{\alpha}_t}\beta_t$. The first equality follows from Bayes' rule, the second from the Markovian property of the forward process, and the third is a rearrangement of Gaussian density functions. Namely, the second loss term $D_{KL}(q(x_{t-1}|x_t, x_0)\|p_\theta(x_{t-1}|x_t))$ from Eq. 7 is also tractable, as it characterizes the KL divergence between two Gaussians and has analytical solutions.

Empirically, Ho et al. (2020) show that predicting the added noise $\epsilon_\theta(x_t, t)$ rather than the mean $\tilde{\mu}_\theta(x_t, t)$ of each forward step leads to a simplified loss function

$$\mathcal{L}_{\text{simple}} := \mathbb{E}_{t,x_0,\epsilon}\left[\|\epsilon - \epsilon_\theta(x_t, t)\|_2^2\right], \tag{9}$$

that performs better in practice. Here, $\epsilon_\theta$ is called a "denoising model", or "denoiser", which predicts the standard normal noise $\epsilon$ added to the input given the input $x_t$ at time step $t$. The denoising model is often formulated as a UNet (Ronneberger et al., 2015) or a Vision Transformer (Dosovitskiy et al., 2020) (see Sec. 4). A more detailed description of how video diffusion models perform such a denoising process can be found in Sec. 5.

To generate samples using DDPM, we utilize the reverse process outlined in Eq. 4 and Eq. 5. This reverse Markov chain allows us to generate a data sample $x_0$ by initially sampling a noise vector $x_T \sim p(x_T)$, and then iteratively sampling from the learnable transition kernel $x_{t-1} \sim p_\theta(x_{t-1} \mid x_t)$ until $t = 1$. The original DDPM generation process has more recently been complemented by a non-Markovian alternative denoted as denoising diffusion implicit models (DDIM, Song et al. 2020), which offers a deterministic and more efficient generation process. Here, a backward denoising step can be computed with

$$\hat{x}_0^{(t)} = \frac{x_t - \sqrt{1 - \overline{\alpha}_t}\epsilon_\theta(x_t, t)}{\sqrt{\overline{\alpha}_t}} \tag{10}$$

$$x_{t-1} = \sqrt{\overline{\alpha}_{t-1}}\hat{x}_0^{(t)} + \sqrt{1 - \overline{\alpha}_{t-1}}\epsilon_\theta(x_t, t). \tag{11}$$

One distinct advantage of DDIM is that it allows for accurate reconstruction of the original input video $x_0$ from the noise at arbitrary time step $t$. To accelerate the generation process, we consider defining the reverse process not on all the intermediate latent variables $x_{1:T}$, but on a subset $\{x_{\tau_1}, \ldots, x_{\tau_S}\}$, where $\tau$ is an increasing subsequence of $[1, \ldots, T]$ of length $S$. Specifically, we define the sequential forward process over $x_{\tau_1}, \ldots, x_{\tau_S}$ such that $q(x_{\tau_i} \mid x_0) = \mathcal{N}(x_{\tau_i}; \sqrt{\overline{\alpha}_{\tau_i}}x_0, (1 - \overline{\alpha}_{\tau_i})\mathbf{I})$ matches the marginals. We can then draw samples following the sequence $x_{\tau_S} \to \hat{x}_0^{(\tau_S)} \to x_{\tau_{S-1}} \to \cdots \to x_{\tau_1}$. This technique, called DDIM inversion, can be utilized for applications such as image and video editing (see Sec. 10).

### 3.2 Score-based Model Formulation

Score-based models (SBMs) (Song & Ermon, 2019; Song et al., 2021) center on the concept of the Stein score, or score function (Hyvärinen, 2005). For a given probability density function $p(x)$, the score function is the gradient of the log probability density, $\nabla_x \log p(x)$. This score function depends on the data $x$ and indicates the direction of the steepest increase in the probability density. SBMs add increasing Gaussian noise to data and estimate the score functions for all noisy data distributions by training a noise-conditional score network (Vincent, 2011). Samples are then generated by chaining these score functions at decreasing noise levels using methods like Langevin dynamics (Parisi, 1981), stochastic differential equations (SDEs) (Song et al., 2021) and ordinary differential equations (ODEs) (Karras et al., 2022). Unlike DDPM, SBMs decouple training and sampling, allowing for various sampling techniques after the score functions are estimated. Below we focus on the two most representative types of SBMs: denoising score matching with Langevin dynamics (SMLD) and score-based stochastic differential equations (Score SDEs).

**Denoising Score Matching with Langevin Dynamics.** In denoising score matching with Langevin dynamics (SMLD), we denote the data distribution by $p(x_0)$ and use a sequence of noise levels $0 < \sigma_1 < \sigma_2 < \cdots < \sigma_T$ for score estimation. A data point $x_0$ is perturbed to $x_t$ by Gaussian noise, $p(x_t \mid x_0) = \mathcal{N}(x_t; x_0, \sigma_t^2\mathbf{I})$. This process creates a series of noisy data distributions $p(x_1), p(x_2), \ldots, p(x_T)$, where $p(x_t) = \int p(x_t \mid x_0)p(x_0)\,\mathrm{d}x_0$. A noise-conditional score network, $s_\theta(x, t)$, is trained to estimate the score function $\nabla_{x_t} \log p(x_t)$ using denoising score matching (Vincent, 2011) with the following training objective

$$\mathbb{E}_{t\sim\mathcal{U}[[1,T]],x_0\sim p(x_0),x_t\sim p(x_t|x_0)}\left[\lambda(t)\|\epsilon + \sigma_t s_\theta(x_t, t)\|^2\right] + \text{const}, \tag{12}$$

where $\mathcal{U}[[1, T]]$ denotes the uniform distribution over discrete indices $\{1, 2, \cdots, T\}$, $x_t = x_0 + \sigma_t\epsilon, \epsilon \sim \mathcal{N}(\mathbf{0}, \mathbf{I})$ and $\lambda(t)$ is a positive weighting function. The training objectives of DDPM and SMLD are equivalent when $\epsilon_\theta(x, t) = -\sigma_t s_\theta(x, t)$.

For sampling, SMLD uses iterative approaches like annealed Langevin dynamics (Song & Ermon, 2019). Starting with $x_T^{(N)} \sim \mathcal{N}(\mathbf{0}, \mathbf{I})$, Langevin dynamics is applied iteratively for $t = T, T-1, \ldots, 1$. At each step, initialized with $x_t^{(0)} = x_{t+1}^{(N)}$, the update rule is:

$$x_t^{(i+1)} = x_t^{(i)} + \frac{1}{2} s_t s_\theta(x_t^{(i)}, t) + \sqrt{s_t} \epsilon^{(i)}, \epsilon^{(i)} \sim \mathcal{N}(\mathbf{0}, \mathbf{I}), \tag{13}$$

where $s_t > 0$ is the step size and $N$ is the number of iterations per step. As $s_t \to 0$ and $N \to \infty$, $x_0^{(N)}$ converges to a valid sample from the data distribution $p(x_0)$.

**Score-based Stochastic Differential Equations.** Models like DDPM and SMLD can be extended to infinite time steps or noise levels using stochastic differential equations (SDEs), forming the Score SDE (Song et al., 2021). This approach uses SDEs for noise perturbation and sample generation, requiring the estimation of score functions of noisy data distributions. Score SDEs perturb data with a forward process governed by:

$$dx = f(x, t) \, dt + g(t) \, dw, \tag{14}$$

where $f(x, t)$ and $g(t)$ are the drift and diffusion functions, and $w$ is a standard Wiener process.

DDPM and SMLD can be seen as discrete versions of this SDE. For DDPM, the corresponding SDE is:

$$dx = -\frac{1}{2}\beta(t)x \, dt + \sqrt{\beta(t)} \, dw, \beta\left(\frac{t}{T}\right) = T\beta_t. \tag{15}$$

For SMLD, the SDE is:

$$dx = \sqrt{\frac{d[\sigma(t)^2]}{dt}} \, dw, \sigma\left(\frac{t}{T}\right) = \sigma_t. \tag{16}$$

Here $T$ refers to the total number of noise levels used in DDPM and SMLD.

Any diffusion process in the form of Eq. 14 can be reversed by solving the reverse-time SDE:

$$dx = \left[f(x, t) - g(t)^2 \nabla_x \log p_t(x)\right] dt + g(t) \, d\bar{w}, \tag{17}$$

where $\bar{w}$ is a reverse-time Wiener process. The solution trajectories of the reverse SDE share the same marginal densities as the forward SDE but evolve in the opposite direction, progressively converting noise into data. Additionally, Song et al. (2021) introduce the probability flow ODE, with trajectories matching those of the reverse-time SDE:

$$dx = \left[f(x, t) - \frac{1}{2}g(t)^2 \nabla_x \log p_t(x)\right] dt. \tag{18}$$

The key to building Score SDE generative models is accurately learning the score function $\nabla_{\mathbf{x}} \log p_t(\mathbf{x})$ at each time step $t$. We achieve this by training a time-dependent score model $\mathbf{s}_\theta(\mathbf{x}_t, t)$ using a continuous-time score matching objective:

$$\mathbb{E}_{t \sim \mathcal{U}[0,T], \mathbf{x}_0 \sim p(\mathbf{x}_0), \mathbf{x}_t \sim p(\mathbf{x}_t | \mathbf{x}_0)} \left[\lambda(t) \|\mathbf{s}_\theta(\mathbf{x}_t, t) - \nabla_{\mathbf{x}_t} \log p(\mathbf{x}_t \mid \mathbf{x}_0)\|^2\right], \tag{19}$$

where $\mathcal{U}[0, T]$ denotes the uniform distribution over $[0, T]$, and other notations follow the definition in Eq. 12.

With a trained score estimation network, we can solve the reverse-time SDE and probability flow ODE to generate samples. Methods include annealed Langevin dynamics Song & Ermon (2019), numerical SDE solvers Song et al. (2021), numerical ODE solvers Song et al. (2021); Karras et al. (2022), and predictor-corrector methods Song et al. (2021), effectively solving the reverse processes.

## 4 Architecture

Next, we review popular architectures used for video diffusion models including UNets and transformers. We first introduce image-based variants before discussing how they may be suitably adapted for video generation, in Sec. 5. We also discuss common variations on these including latent diffusion models and cascaded diffusion models.

### 4.1 UNet

The UNet (Ronneberger et al., 2015) is currently the most popular architectural choice for the denoiser in visual diffusion models (see Figure 4). Originally developed for medical image segmentation, it has more recently been successfully adapted for generative tasks in image, video, and audio domains. A UNet transforms an input image into an output image of the same size and shape by encoding the input first into increasingly lower spatial resolution latent representations while increasing the number of feature channels by progressing through a fixed number of encoding layers. Then, the resulting 'middle' latent representation is upsampled back to its original size through the same number of decoding layers. While the original UNet (Ronneberger et al., 2015) only used ResNet blocks, most diffusion models interleave them with Vision Transformer blocks in each layer. The ResNet blocks mainly utilize 2D-Convolutions, while the Vision Transformer blocks implement spatial self-attention, as well as cross-attention. This happens in a way that allows conditioning of the generative process on additional information such as text prompts and current timestep. Layers of the same resolution in the encoder and decoder part of the UNet are connected through residual connections. The UNet can be trained by the process outlined in Sec. 3.

### 4.2 Vision Transformer

The Vision Transformer (ViT, Dosovitskiy et al. (2020)) is an important building block of generative diffusion models based on the transformer architecture developed for natural language processing (Vaswani

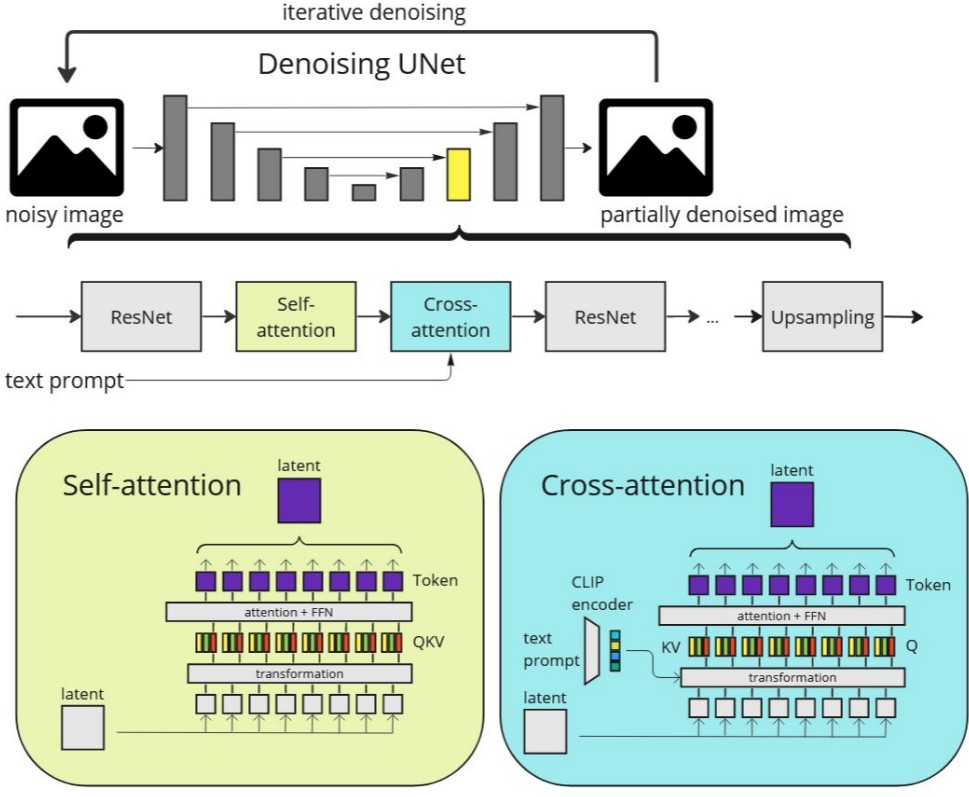

Figure 4: The denoising UNet architecture typically used in text-to-image diffusion models. The model iteratively predicts a denoised version of the noisy input image. The image is processed through a number of encoding layers and the same number of decoding layers that are linked through residual connections. Each layer consists of ResNet blocks implementing convolutions, as well as Vision Transformer self-attention and cross-attention blocks. Self-attention shares information across image patches, while cross-attention conditions the denoising process on text prompts.

et al., 2017). Therefore, it similarly combines multi-head attention layers, normalization layers, residual connections, as well as a linear projection layer to transform a vector of input tokens into a vector of output tokens. In the image case, the input tokens are obtained by dividing the input image into regular patches and using an image encoder to compute for each patch a patch embedding, supplemented with learnable position embeddings. Within the attention layer, the patch embeddings are projected through trainable projection matrices, producing so called Query, Key and Value matrices. The first two matrices are used to compute a learnable affinity matrix A between different image token positions, which is calculated according to the scaled dot-product attention formula: $A(Q, K) = \text{softmax}(\frac{QK^T}{\sqrt{d_k}})$. Here, $Q$ and $K$ are $d \times d_k$ dimensional and refer to the query and key matrix, $d$ is the number of input tokens, $d_k$ the dimensionalities of the $d$ query and key vectors making up the rows of $K$ and $Q$, and the matrix $Z$ of output embeddings is obtained as $Z = AV$, i.e. the attention-weighted superposition of the rows of the value matrix $V$ (with one row for each input token embedding). In the simplest case, there is a single ($d \times d$ dimensional) affinity matrix, resulting from a single set of projection matrices. In multi-head attention, a stack of such projections with separate query, key, and value matrices is used. Their outputs are concatenated and transformed through a linear output layer to form a single set of $d$ new patch embeddings. The attention heads can be computed in parallel and allow the model to focus on multiple aspects of the image. Depending on the task, ViTs can output an image embedding or be equipped with a classification head.

In diffusion models, ViT blocks serve two purposes: On the one hand, they implement spatial self-attention where $Q$, $K$, and $V$ refer to image patches. This allows information to be shared across the whole image, or even an entire video sequence. On the other hand, they are used for cross-attention that conditions the denoising process on additional guiding information such as text prompts. Here, $Q$ is an image patch and $K$ and $V$ are based on text tokens that have been encoded into an image-like representation using a CLIP encoder (Radford et al., 2021).

Purely Vision Transformer-based diffusion models have been proposed as an alternative to the standard UNet (Peebles & Xie, 2022; Lu et al., 2023b; Ma et al., 2024; Chen et al., 2023c;b; Gupta et al., 2023). Rather than utilizing convolutions, the whole model consists of a series of transformer blocks only. This approach has distinct advantages, such as flexibility in regard to the length of the generated videos. While UNet-based models typically generate output sequences of a fixed length, transformer models can auto-regressively predict tokens in sequences of relatively arbitrary length.

### 4.3 Cascaded Diffusion Models

Cascaded Diffusion Models (CDM, Ho et al. 2022b) consist of multiple UNet models that operate at increasing image resolutions. By upsampling the low-resolution output image of one model and passing it as input to the next model, a high-fidelity version of the image can be generated. At training time, various forms of data augmentation are applied to the outputs of one denoising UNet model before it is passed as input to the next model in the cascade. These include Gaussian blurring, as well as premature stopping of the denoising process (Ho et al., 2022b). The use of CDMs has largely vanished after the adaptation of Latent Diffusion Models (Rombach et al., 2022) that allow for native generation of high-fidelity images with lower resources.

### 4.4 Latent Diffusion Models

Latent Diffusion Models (LDM, Rombach et al. (2022)) have been an important development of the base UNet architecture that now forms the de-facto standard for image and video generation tasks. Instead of operating in RGB space, the input image is first encoded into a latent representation with lower spatial resolution and more feature channels using a pre-trained vector-quantized variational auto-encoder (VQ-VAE, Van Den Oord et al. (2017)). This low-resolution representation is then passed to the UNet where the whole diffusion and denoising process takes place in the latent space of the VQ-VAE encoder. The denoised latent is then decoded back to the original pixel space using the decoder part of the VQ-VAE. By operating in a lower-dimensional latent space, LDMs can save significant computational resources, thus allowing them to generate higher-resolution images compared to previous diffusion models. Stable Diffusion [1] is a canonical

---

[1]https://github.com/Stability-AI/stablediffusion

open source implementation of the LDM architecture. Further improvements of the LDM architecture have been introduced by Chen et al. (2020), who addressed specific concerns about how to adjust the architecture for high-resolution images, and Podell et al. (2023), who used a second refiner network for improving the sample quality of generated images.

## 5    Temporal Dynamics

In diffusion-based video generation, the UNet/ViT models described in Sec. 4 are used as denoising models ($\epsilon_\theta$ in Eq. 9) to predict the noise added to the input video clip. Unlike image diffusion models that generate each image separately, video diffusion models often perform denoising over a set of frames simultaneously. In other words, the input $x_t$ to the video diffusion model represents an $n$-frame video clip (e.g. $n = 16$). Formally, given a noisy video clip (or video latent if using latent diffusion models) $x_t$ at time step $t$, the UNet/ViT model predicts the noise $\epsilon$ added to the video clip/latent, which is then used to derive a less noisy version of the video $x_{t-1}$. This denoising process can be repeated until the clean video clip/latent $x_0$ is obtained.

Text-to-image models such as Stable Diffusion can produce realistic images, but extending them for video generation tasks is not trivial (Ho et al., 2022c). If we try to naively generate individual video frames from a text prompt, the resulting sequence has no spatial or temporal coherence (see Figure 5a). For video editing tasks, we can extract spatial cues from the original video sequence and use them to condition the diffusion process. In this way, we can produce fluid motion of objects, but temporal coherence still suffers due to changes in the finer texture of objects (see Figure 5b). In order to achieve spatio-temporal consistency, video diffusion models need to share information across video frames. The most obvious way to achieve this is to add a third temporal dimension to the denoising model. ResNet blocks then implement 3D convolutions, while self-attention blocks are turned into full cross-frame attention blocks. This type of full 3D architecture is however associated with very high computational costs.

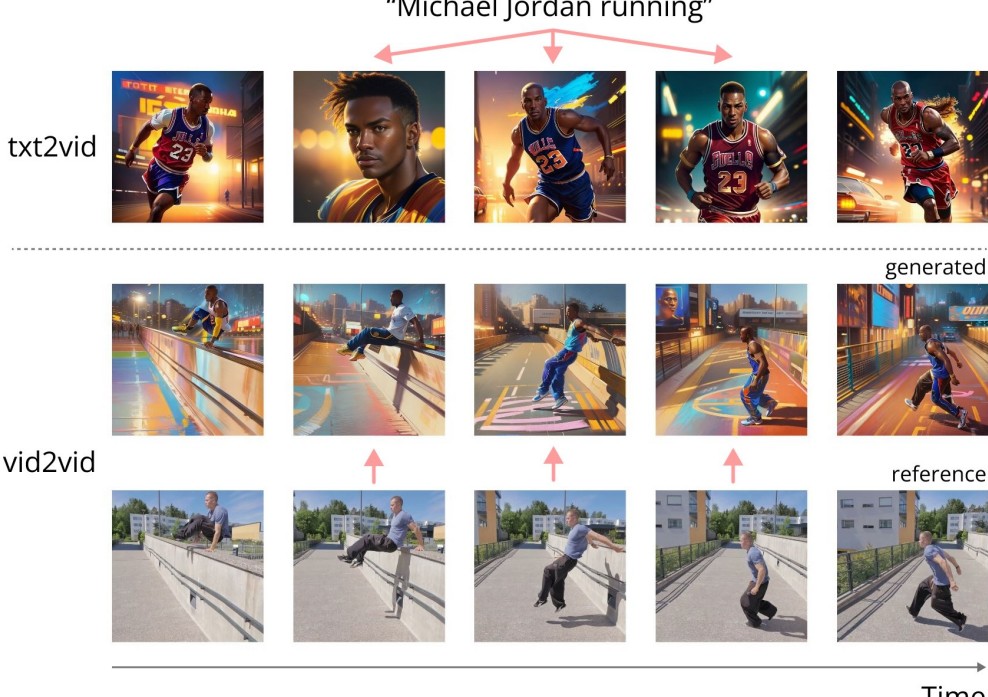

Figure 5: Limitations of text-to-video diffusion models for generating consistent videos. (Top) When using only a text prompt ("Michael Jordan running"), both the appearance and position of objects change wildly between video frames. (Bottom) Conditioning on spatial information from a reference video can produce consistent movement, but the appearance of objects and the background still fluctuate between video frames.

To lower the computational demands of video UNet models, different approaches have been proposed (see Figure 6 and Figure 7): 3D convolution and attention blocks can be factorized into spatial 2D and temporal 1D blocks. The temporal 1D modules are often inserted into a pre-trained text-to-image model. Additionally, temporal upsampling techniques are often used to increase motion consistency. In video-to-video tasks, pre-processed video features such as depth estimates are often used to guide the denoising process. Finally, the type of training data and training strategy has a profound impact on a model's ability to generate consistent motion.

### 5.1 Spatio-Temporal Attention Mechanisms

In order to achieve spatial and temporal consistency across video frames, most video diffusion models modify the self-attention layers in the UNet model. These layers consist of a vision transformer that computes the affinity between a query patch of an image and all other patches in that same image. This basic mechanism can be extended in several ways (see Wang et al. 2023b for a discussion): In temporal attention (Hong et al., 2022; Singer et al., 2022), the query patch attends to patches at the same location in other video frames. In full spatio-temporal attention (Zhang & Agrawala, 2023; Bar-Tal et al., 2024), it attends to all patches in all video frames. In causal attention, it only attends to patches in all previous video frames. In sparse causal attention (Wu et al., 2022b), it only attends to patches in a limited number of previous frames, typically the first and immediately preceding one. The different forms of spatio-temporal attention differ in how computationally demanding they are and how well they can capture motion. Additionally, the quality of the produced motion greatly depends on the used training strategy and data set.

### 5.2 Temporal Upsampling

Generating long video sequences in a single batch often exceeds the capacity of current hardware. While different techniques have been explored to reduce the computational burden (such as sparse causal attention, Wu et al. 2022b), most models are still limited to generating video sequences that are no longer than a few seconds even on high-end GPUs. To get around this limitation, many works have adapted a hierarchical upsampling technique whereby they first generate spaced-out key frames. The intermediate frames can then be filled in by either interpolating between neighboring key frames, or using additional passes of the diffusion model conditioned on two key frames each.

As an alternative to temporal upsampling, the generated video sequence can also be extended in an auto-regressive manner (Blattmann et al., 2023b). Hereby, the last generated video frame(s) of the previous batch are used as conditioning for the first frame(s) of the next batch. While it is in principle possible to arbitrarily extend a video in this way, the results often suffer from repetition and quality degradation over time.

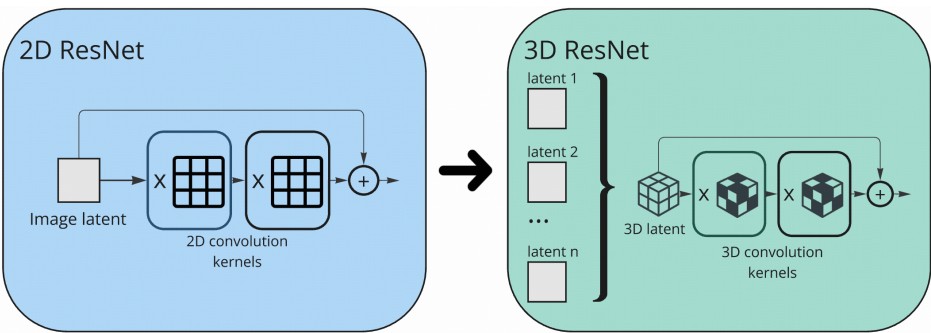

Figure 6: Three-dimensional extension of the UNet convolution layers for video generation. The convolution kernels are expanded into 3D kernels to adapt to the extra temporal dimension in video generation.

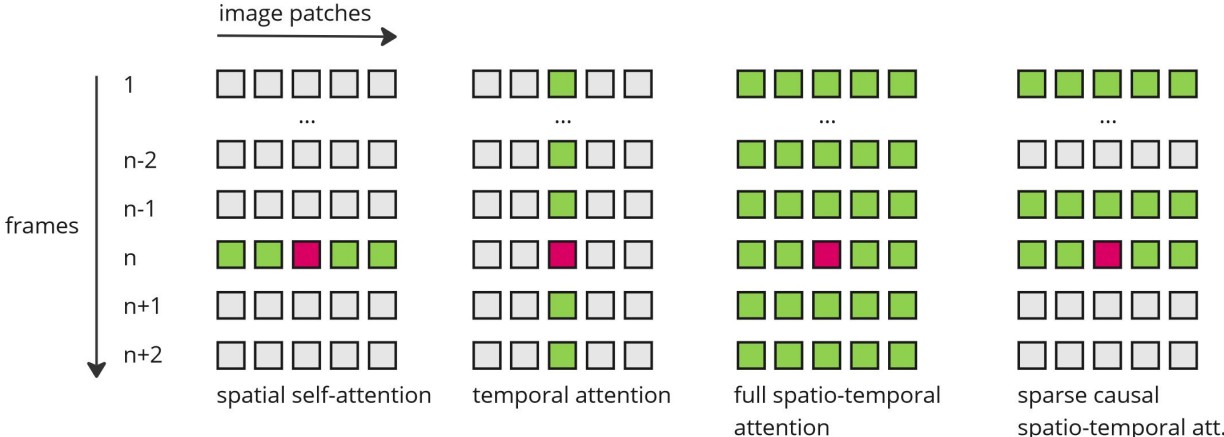

Figure 7: Attention mechanisms used to model temporal dynamics are illustrated here. The color red represents the Query, and green indicates the Key. This diagram identifies which patches in the 3D video sequences receive attention.

### 5.3 Structure Preservation

Video-to-video translation tasks typically strive for two opposing objectives: Maintaining the coarse structure of the source video on the one hand, while introducing desired changes on the other hand. Adhering to the source video too much can hamper a model's ability to perform edits, while strolling too far away from the layout of the source video allows for more creative results but negatively impacts spatial and temporal coherence.

A common approach for preserving the coarse structure of the input video is to replace the initial noise in the denoising model with (a latent representation of) the input video frames (Wu et al., 2022b). By varying the amount of noise added to each input frame, the user can control how closely the output video should resemble the input, or how much freedom should be granted while editing it. In practice, this method in itself is not sufficient for preserving the more fine-grained structure of the input video and is therefore usually augmented with other techniques. For one, the outlines of objects are not sufficiently preserved when adding higher amounts of noise. This can lead to unwanted object warping across the video. Furthermore, finer details can shift over time if information is not shared across frames during the denoising process.

These shortcomings can be mitigated to some degree by conditioning the denoising process on additional spatial cues extracted from the original video. For instance, specialized diffusion models have been trained to take into account depth estimates[2]. ControlNet (Zhang & Agrawala, 2023) is a more general extension for Stable Diffusion that enables conditioning on various kinds of information, such as depth maps, OpenPose skeletons, or lineart. A ControlNet model is a fine-tuned copy of the encoder portion of the Stable Diffusion denoising UNet that can be interfaced with a pre-trained Stable Diffusion model. Image features are extracted using a preprocessor, encoded through a specialized encoder, passed through the ControlNet model, and concatenated with the image latents to condition the denoising process. Multiple ControlNets can be combined in an arbitrary fashion. Several video diffusion models have also implemented video editing that is conditioned on extracted frame features such as depth (Ceylan et al. 2023; Esser et al. 2023; Xing et al. 2023a, see Sec. 10.2) or pose estimates (Ma et al. 2023; Zhao et al. 2023, see Sec. 10.3).

## 6 Training & Evaluation

Video diffusion models can differ greatly in regards to how they are trained. Some models are trained from scratch (e.g. Ho et al. 2022c, Singer et al. 2022, Ho et al. 2022a), while others are built on top of a pre-

---

[2]https://huggingface.co/stabilityai/stable-diffusion-2-depth

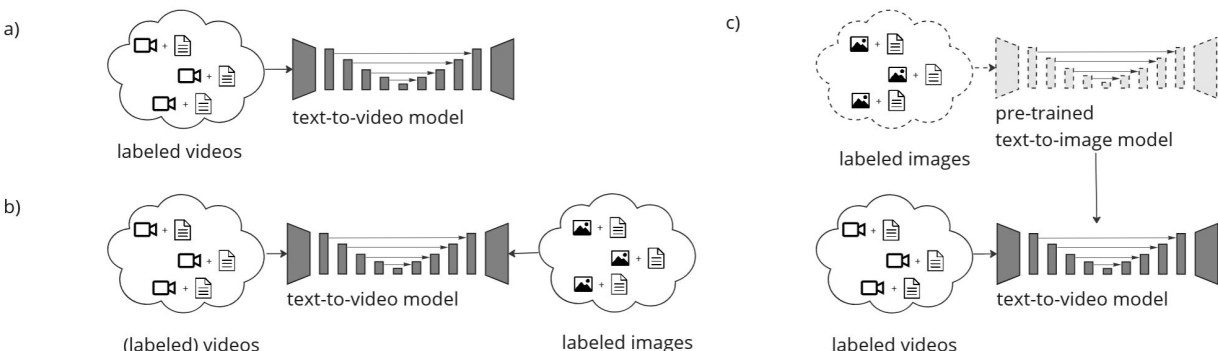

Figure 8: Training approaches for video diffusion models: a) Training on videos. b) Simultaneous training on images and videos. c) Pre-training on images and fine-tuning on videos.

Table 2: Commonly used video data sets for diffusion model training.

| Data Set | Resolution | Source | Labels | # Clips | Clip Length | Total Length (h.) |
|---|---|---|---|---|---|---|
| WebVid-10M (2021) | - | Web | Alt-Text | 10.7 mio. | 18 sec. | 52k |
| HD-Vila-100M (2022) | 1280×720 | Youtube | Transcription (auto.) | 100 mio. | 13.4 sec. | 371k |
| Kinetics-600 (2018) | - | Youtube | Action Class | 500,000 | 10 sec. | 1.4k |
| UCF101 (2012) | 320×240 | Youtube | Action Class | 13k | 7 sec. | 27 |
| MSR-VTT (2016) | - | Web | Annotation (human) | 10k | 10-30 sec. | 41.2 |
| Sky Time-lapse (2018) | 640×360 | Youtube | - | 35k | 32 frames | 10 |
| Tai-Chi-HD (2019) | 256×256 | Youtube | - | 3k | 128-1024 frames | - |
| TikTok (2022) | 640×1080 | TikTok | Depth | 340 | 10-15 sec. | 0.86 |

trained image model (e.g. Zhou et al. 2022, Khachatryan et al. 2023, Blattmann et al. 2023b). It is possible to train a model completely on labeled video data, whereby it learns associations between text prompts and video contents as well as temporal correspondence across video frames (e.g. Ho et al. 2022c). However, large data sets of labeled videos (e.g. Bain et al. 2021, Xue et al. 2022, see Section 6.1) tend to be smaller than pure image data sets and may include only a limited range of content. Additionally, a single text label per video may fail to describe the changing image content across all frames. At a minimum, automatically collected videos need to be divided into chunks of suitable length that can be described with a single text annotation and that are free of unwanted scene transitions, thereby posing higher barriers for uncurated or weakly curated data collection. For that reason, training is often augmented with readily available data sets of labeled images (e.g. Russakovsky et al. 2015, Schuhmann et al. 2022, see Section 6.2). This allows a given model to learn a broader number of relationships between text and visual concepts. Meanwhile, the spatial and temporal coherence across frames can be trained independently on video data that can even be unlabeled (Zhou et al., 2022).

In contrast to models that are trained from scratch (e.g. Ho et al. 2022c, Singer et al. 2022, Ho et al. 2022a), recent video diffusion approaches (e.g. Zhou et al. 2022, Khachatryan et al. 2023, Blattmann et al. 2023b) often rely on a pre-trained image generation model such as Stable Diffusion (Rombach et al. 2022). These models show impressive results in the text-to-image (Rombach et al., 2022; Ramesh et al., 2022) and image editing domains (Brooks et al., 2023; Zhang & Agrawala, 2023), but are not built with video generation in mind. For this reason, they have to be adjusted in order to yield results that are spatially and temporally coherent. One possibility to achieve this is to add new attention blocks or to tweak existing ones so that they model the spatio-temporal correspondence across frames. Depending on the implementation, these attention blocks either re-use parameters from the pre-trained model, are fine-tuned on a training data set consisting of many videos, or only on a single input video in the case of video-to-video translation tasks. During fine-tuning, the rest of the pre-trained model's parameters are usually frozen in place. The different training methods are shown in Figure 8.

### 6.1 Video Data Sets

Table 2 offers an overview of commonly used video data sets for training and evaluation of video diffusion models.

**WebVid-10M** (Bain et al., 2021) is a large data set of text-video pairs scraped from the internet that covers a wide range of content. It consists of 10.7 million video clips with a total length of about 52,000 hours. It is an expanded version of the WebVid-2M data set, which includes 2.5 million videos with an average length of 18 seconds and a total play time of 13,000 hours. Each video is annotated with an HTML Alt-text which normally serves the purpose of making it accessible to vision-impaired users. The videos and their Alt-texts have been selected based on a filtering pipeline similar to that proposed in Sharma et al. (2018). This ensures that the videos have sufficiently high resolution, normal aspect ratio, and lack profanity. Additionally, only well-formed Alt-text that is aligned with the video content is selected (as judged by a classifier). WebVid-10M is only distributed in the form of links to the original video sources, therefore it is possible that individual videos that have been taken down by their owners are no longer accessible.

**HD-Villa-100M** (Xue et al., 2022) contains over 100 million short video clips extracted from about 3.3 million videos found on YouTube. The average length of a clip is 13.4 seconds with a total run time of about 371.5 thousand hours. All videos have a high-definition resolution of $1280 \times 720$ pixels and are paired with automatic text transcriptions. Along with WebVid-10M, HD-Villa-100M is one of the most popular training data sets for generative video models.

**Kinetics-600** (Carreira et al., 2018) contains short YouTube videos of 600 distinct human actions with their associated class labels. Each action class is represented by more than 600 video clips that last around 10 seconds. In total, the data set contains around 500,000 clips. The data set expands upon the previous Kinetics-400 (Kay et al., 2017) data set.

**UCF101** (Soomro et al., 2012) is a data set of videos showing human actions. It contains over 13,000 YouTube video clips with a total duration of about 27 hours and an average length of 7 seconds. It expands upon the previous UCF50 (Reddy & Shah, 2013) data set, which includes only roughly half as many video clips and action classes. The clips have a resolution of $320 \times 240$ pixels. Each video has been annotated with a class label that identifies it as showing one of 101 possible actions. The 101 action classes are more broadly categorized into 5 action types: Human-Object Interaction, Body-Motion Only, Human-Human Interaction, Playing Musical Instruments, and Sports. While UCF101 was mainly intended for training and evaluating action classifiers, it has also been adopted as a benchmark for generative models. For this, the class labels are often used as text prompts. The generated videos are then usually evaluated using IS, FID, and FVD metrics.

**MSR-VTT** (Xu et al., 2016) includes about 10,000 short video clips from over 7,000 videos with a total run time of about 41 hours. The videos were retrieved based on popular video search queries and filtered according to quality criteria such as resolution and length. Each clip was annotated by 20 different humans with a short text description, yielding 200,000 video-text pairs. The data set was originally intended as a benchmark for automatic video annotation but has been used for evaluating text-to-video models as well. For this, CLIP text-similarity, FID, and FVD scores are usually reported.

**Sky Time-lapse** (Xiong et al., 2018) is a collection of unlabeled short clips that contain time-lapse shots of the sky. The videos have been taken from YouTube and divided into smaller non-overlapping segments. Each clip consists of 32 frames of continuous video at a resolution of $640 \times 360$ pixels. The clips show the sky at different times of day, under different weather conditions, and with different scenery in the background. The data set can serve as a benchmark for unconditional video generation or video prediction. In particular, it allows one to assess how well a given generative video model is able to replicate complex motion patterns of clouds and stars.

**Tai-Chi-HD** (Siarohin et al., 2019) contains over 3,000 unlabeled clips from 280 tai chi Youtube videos. The videos have been split into smaller chunks that range from 128 to 1024 frames and have a resolution of $256 \times 256$ pixels. Similar to Sky Time-lapse, Tai-Chi-HD can be used for training and evaluating unconditional generation or video prediction.

**TikTok Dataset** (Jafarian & Park, 2022) is recognized as a popular benchmark for the generation of human dancing videos. It comprises approximately 350 dance videos, each lasting between 10 and 15 seconds. The curated videos in the dataset capture a single individual performing dance moves from monthly TikTok dance challenge compilations. These selected videos showcase moderate movements without significant motion blur. For each video, RGB images are extracted at a rate of 30 frames per second, resulting in a total of more than 100,000 images. Human poses, depth maps, segmentation masks, and UV coordinates of each video are also provided.

## 6.2 Image Data Sets

Video models are sometimes jointly trained on image and video data. Alternatively, they may extend a pre-trained image generation model with temporal components that are fine-tuned on videos. Table 3 provides a brief overview over commonly used labeled image data sets.

**ImageNet** (Russakovsky et al., 2015) is a data set developed for the ImageNet Large Scale Visual Recognition Challenge that was held annually between 2010 and 2017. Since 2012, the same data set has been used for the main image classification task. ImageNet-21k is a large collection of over 14 million images that have been annotated by humans with one object category label. Overall, there are 20,000 different object classes present in the data set that are hierarchically organized according to the WordNet (Fellbaum, 1998) structure. A subset of this dataset used for the ImageNet competition itself is often called ImageNet-1k. It contains over 1 million images that each have been annotated by humans with one object category label and a corresponding bounding box. There are only 1,000 object categories in this data set.

**MS-COCO** (Lin et al., 2014) has originally been developed as a benchmark data set for object localization models. It contains over 300,000 images containing 91 different categories of everyday objects. Every instance of an object is labeled with a segmentation mask and a corresponding class label. Overall, there are about 2.5 million instances of objects in this data set.

**LAION-5B** (Schuhmann et al., 2022) is a very large public collection of 5.58 billion text-image pairs that can be found on the internet. Access is provided in the form of a list of links. To ensure a minimal level of correspondence between the images and their associated alt-texts, the pairs have been filtered by the following method: Images and texts have both been encoded through a pre-trained CLIP (Radford et al., 2021) model and pairs with a low cosine CLIP similarity have been excluded. To train image or video models, often only the subset of LAION-5B that contains English captions is used. It contains 2.32 billion text-image pairs and is referred to as LAION-2B. Additionally, labels for *not safe for work* (NSFW), watermarked, or toxic content are provided based on automated classification. The LAION-5B data set offers are relatively low level of curation, but its sheer size has proven very valuable for training large image and video models.

## 6.3 Evaluation Metrics

**Human ratings** are the most important evaluation method for video models since the ultimate goal is to produce results that appeal to our aesthetic standards. To demonstrate the quality of a new model, subjects usually rate its output in comparison to an existing baseline. Subjects are usually presented with pairs of generated clips from two different video models. They are then asked to indicate which of the two examples they prefer in regard to a specific evaluation criterion. Depending on the study, the ratings can either purely reflect the subject's personal preference, or they can refer to specific aspects of the video such as temporal consistency and adherence to the prompt. Humans are very good at judging what "looks natural"

Table 3: Commonly used image data sets that are used for video diffusion model training.

| Data Set | # Images | Annotation | Labels | # Classes |
|---|---|---|---|---|
| ImageNet-21k (2015) | 14 mio. | Human | Class | 20,000 |
| ImageNet-1k (2015) | 1.28 mio. | Human | Class | 1,000 |
| MS-COCO (2014) | 328k | Human | Class | 91 |
| LAION-5B (2022) | 5.58 mio. | Automated | Text | - |

and identifying small temporal inconsistencies. The downsides of human ratings include the effort and time needed to collect large enough samples, as well as the limited comparability across studies. For this reason, it is desirable to also report automated evaluation metrics. Human studies can also be used to measure how well the automated metrics align with human preferences, checking if human judgments agree with metric results or differ when assessing similar videos (Unterthiner et al., 2018; Huang et al., 2024; Liu et al., 2024).

Common automated evaluation metrics can be categorized into two types: 1) set-to-set comparison metrics, and 2) unary metrics, as shown in Figure 9. The first category measures the difference between the generated set of data and the reference dataset, typically using statistical measures such as Fréchet distance (Dowson & Landau, 1982). The second category, unary metrics, does not require a reference set. This makes them suitable for applications like video generation in the wild or video editing, where a gold-standard reference is absent.

### 6.3.1 Set-to-set Comparison Metrics

**Fréchet Inception Distance** (FID, Heusel et al. 2017) measures the similarity between the output distribution of a generative image model and its training data. Rather than comparing the images directly, they are first encoded by a pre-trained inception network (Szegedy et al., 2016). The FID score is calculated as the squared Wasserstein distance between the image embeddings in the real and synthetic data. FID can be applied to individual frames in a video sequence to study the image quality of generative video models, but it fails to properly measure temporal coherence.

**Fréchet Video Distance** (FVD, Unterthiner et al. 2018) has been proposed as an extension of FID for the video domain. Its inception net is comprised of a 3D Convnet pre-trained on action recognition tasks in YouTube videos (I3D, Carreira & Zisserman 2017). The authors demonstrate that the FVD measure is not only sensitive to spatial degradation (different kinds of noise), but also to temporal aberrations such as swapping of video frames. FVD is a commonly used metric for assessing the quality of unconditional or text-conditioned video generation.

**Kernel Video Distance** (KVD, Unterthiner et al. 2018) is an alternative to FVD. It is computed in an analogous manner, except that a polynomial kernel is applied to the features of the inception net. The authors found that FVD aligns better with human judgments than KVD. Nevertheless, both are commonly reported as benchmark metrics for unconditional video generation.

**Fréchet Video Motion Distance** (FVMD, Liu et al. 2024) is a metric focused on temporal consistency, measuring the similarity between motion features of generated and reference videos using Fréchet Distance. It begins by tracking keypoints using the pre-trained PIPs++ model (Zheng et al., 2023), then calculates the velocity and acceleration fields for each frame. The metric aggregates these features into statistical histograms and measures their differences using the Fréchet Distance. FVMD assesses motion consistency by analyzing speed and acceleration patterns, assuming smooth motions should follow physical laws and avoid abrupt changes.

In addition to video-based metrics, the **Peak Signal-to-Noise Ratio (PSNR)** and **Structural Similarity Index Measure (SSIM)** (Wang et al., 2004) are commonly used image-level metrics for video quality assessment. Specifically, SSIM characterizes the brightness, contrast, and structural attributes of the reference and generated videos, while PSNR quantifies the ratio of the peak signal to the Mean Squared Error

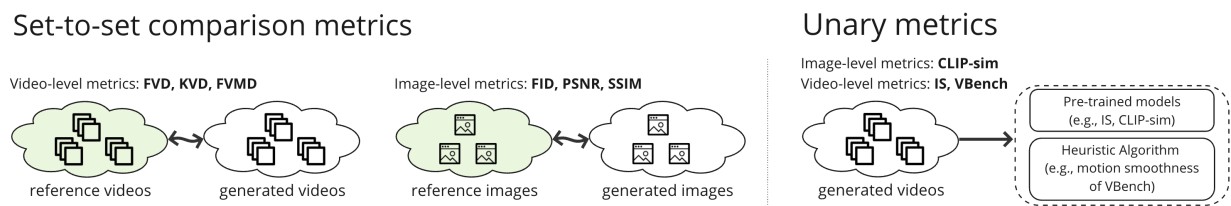

Figure 9: Commonly used algorithmic video evaluation metrics.

(MSE). Originally proposed for imaging tasks such as super-resolution and in-painting, these metrics are nonetheless repurposed for video evaluation. Unlike the aforementioned methods, PSNR and SSIM do not need pre-trained models.

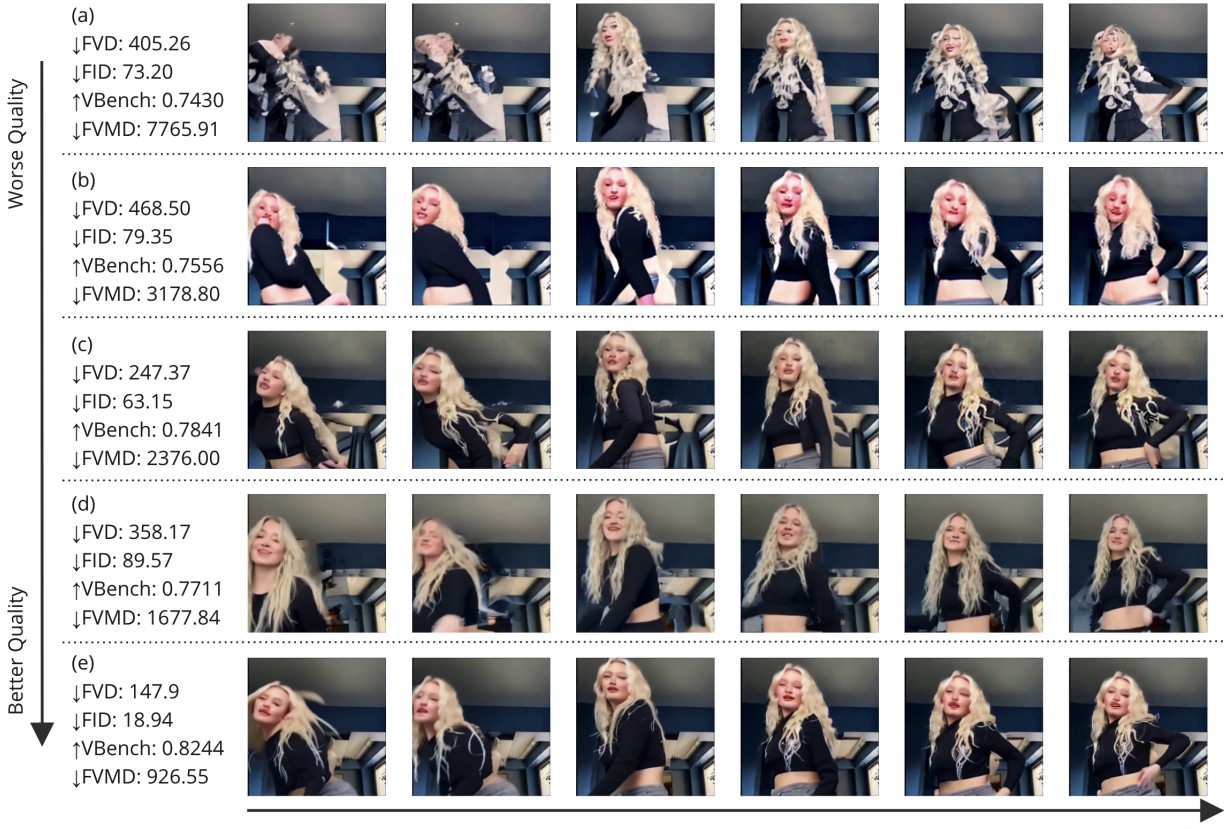

Figure 10: Limitations of Automated Evaluation Metrics: We evaluate various video generative models trained on the TikTok dataset (Jafarian & Park, 2022) to compare their sample quality. Specifically, we generate about 50 videos for each model checkpoint and measure their performance using the FVD (Unterthiner et al., 2018), FID (Heusel et al., 2017), VBench (Huang et al., 2024), and FVMD (Liu et al., 2024) metrics. We present visualizations of five video samples from the same scene, ranked by human assessors from worst to best, to highlight the limitations of algorithmic evaluation methods. The video samples are reproduced from the following models: (a) is from Magic Animate (Xu et al., 2023); (b), (c), and (e) are from Animate Anyone (Hu et al., 2023b), each with different training hyperparameters; and (d) is from DisCo (Wang et al., 2023a). For instance, video sample (a), which is of the poorest quality, cannot be effectively distinguished from samples (b), (c), and (d) based on the FID or VBench metrics. In contrast, the FVMD metric aligns better with the assessed video quality and motion consistency.

### 6.3.2 Unary Metrics

**Inception Score** (IS, Salimans et al. 2016) is applicable to generative models trained on data sets with categorical labels. An Inception Net (Szegedy et al., 2016) classifier pre-trained on the ImageNet data set (Deng et al., 2009) is used to predict the class labels of each generated image. The IS score is then expressed by the Kullback-Leibler distance between the conditional class probability distribution $p(y|G(x))$ and the marginal class distribution $p(y)$ of the generated samples. While IS aligns well with human ratings and possesses good discriminative power (Borji, 2019), it is susceptible to noise, as shown by Heusel et al. (2017). It should be noted that IS only assesses the quality of individual images. When applied to video data,

it can therefore not take into account aspects such as temporal coherence between video frames. Saito et al. (2020) generalizes the IS to the video domain, specifically for the UCF101 dataset (Soomro et al., 2012), where a pre-trained action recognition classifier (C3D, Tran et al. 2015) is used for score computation. However, this metric is generally highly specific to the UCF101 dataset and is hardly applicable to videos in the wild due to classification difficulty.

**CLIP cosine similarity** is often used to measure text prompt and frame consistency, where a reference video dataset is not needed. CLIP (Radford et al., 2021) is a family of vision transformer auto-encoder models that can project image and text data into a shared embedding space. During training, the distance between embedded images and their associated text labels is minimized. Thereby, visual concepts are represented close to words that describe them. The similarity between CLIP embeddings is typically measured through their cosine distance. A value of 1 describes identical concepts, while a value of 0 implies completely unrelated concepts. In order to determine how well a video sequence adheres to the text prompt used to generate or edit it, the average similarity between each video frame and the text prompt is calculated (prompt consistency, Esser et al. 2023). In a similar fashion, it is also possible to get a rough measure of temporal coherence by computing the mean CLIP similarity between adjacent video frames in a sequence (frame consistency, Esser et al. 2023). In video editing tasks, the percentage of frames with a higher prompt consistency score in the edited over the original video is also sometimes reported (frame accuracy, Qi et al. 2023).

**VBench** (Huang et al., 2024) proposes a comprehensive set of fine-grained video evaluation metrics to assess temporal and frame-wise video quality, as well as video-text consistency in terms of semantics and style. They employ a number of pre-trained models, *e.g.*, RAFT (Teed & Deng, 2020) for dynamic degree, and MUSIQ (Ke et al., 2021) for imaging quality, along with heuristics-inspired algorithms, *e.g.*, visual smoothness and temporal flickering, based on inter-frame interpolation and reconstruction error. The overall score is determined by a weighted sum of a number of fine-grained metrics, and the authors also conduct human studies to validate the effectiveness of these metrics.

## 6.4 Benchmarks

Some authors train their video models on specific datasets for evaluation purposes. This allows them to directly compare them with earlier models that have been trained on the same data. In this way, certain datasets can also be seen as quality benchmarks for new video diffusion models. Commonly used evaluation datasets for video generation include UCF-101 (Soomro et al., 2012), MSR-VTT (Xu et al., 2016), Tai-Chi-HD (Siarohin et al., 2019), and Sky Time-Lapse (Radford et al., 2021). Models trained on all four of these datasets can be evaluated using samples that have been generated in an unconditional manner. For UCF-101, a second benchmark is sometimes reported on conditional generation where the 101 class labels are used for guiding the generative process. In this case, IS can be used as an evaluation metric. For MSR-VTT, conditional generation using the 200,000 human video annotations as text prompts can also be evaluated. Here, CLIP text-similarity is often reported as a measure of text-video alignment. Most often, the benchmarked models are either directly trained on the train split of the evaluation data set, or they are pre-trained on a separate large video data set (such as WebVid-10M or HD-Villa-100M) and later fine-tuned on the evaluation data set. However, some papers evaluate their model in a zero-shot setting, where the model has not been trained on the evaluation data set at all. These discrepancies between evaluation setups mean that a direct comparison of benchmark results across studies should be taken with a grain of salt. More complex benchmarking suites include Ego-Exo4D (Grauman et al., 2024) which is a multimodal and multiview video dataset of skilled human activities.

The benchmark results for video generation are summarized in Table 4. Make-A-Video (Singer et al., 2022), one of the early diffusion-based video models, still holds state-of-the-art FVD and IS scores on the UCF-101 conditional generation benchmark. It not only outperforms all GAN and auto-regressive models, but also the newer diffusion-based models. It is pre-trained on both the WebVid-10M and HD-Villa-100M data sets, which gives it an advantage in terms of the quantity of the training data over most other models. Make-A-Video also holds the best CLIP-similarity and FID scores in the zero-shot text-conditioned MSR-VTT benchmark. Make-A-Video is outperformed by Make-Your-Video (Xing et al., 2023a) when it comes to zero-shot performance on UCF-101, although the latter uses depth maps as additional conditioning. Therefore, both models are not directly comparable. VideoFusion (Luo et al., 2023) has achieved the best FVD score

on the unconditional Tai-Chi-HD and Sky Time-lapse benchmarks, as well as the best KVD score on Tai-Chi-HD. It is outperformed by LVDM (He et al., 2022b) when it comes to KVD on the Sky Time-lapse benchmark. MagicVideo (Zhou et al., 2022) has the best FID score on UCF-101 and the best FVD score on MSR-VTT, although our comparison includes only few other datasets competing in those categories.

# 7 Video Generation

## 7.1 Unconditional Generation & Text-to-Video

Unconditional video generation and text-conditioned video generation are common benchmarks for generative video models. Prior to diffusion models, Generative Adversarial Networks (GANs, Goodfellow et al. 2014, Melnik et al. 2024) and auto-regressive transformer models (Vaswani et al., 2017) have been popular choices for generative video tasks. In the following, we provide a short overview over a few representative GAN and auto-regressive transformer video models. We then introduce a selection of competing diffusion models starting in Sec. 7.1.3.

### 7.1.1 GAN Video Models

**TGAN** (Saito et al., 2017) employs two generator networks: The temporal generator creates latent features that represent the motion trajectory of a video. This feature vector can be fed into an image generator that creates a fixed number of video frames in pixel space. TGAN-v2 (Saito et al., 2020) uses a cascade of generator modules to create videos at various temporal resolutions, making the process more efficient. TGAN-F (Kahembwe & Ramamoorthy, 2020) is another improved version that relies on lower-dimensional kernels in the discriminator network.

**MoCoGAN** (Tulyakov et al., 2018) decomposes latent space into motion and content-specific parts by employing two separate discriminators for individual frames and video sequences. At inference time, the content vector is kept fixed while the next motion vector for each frame is predicted in an auto-regressive manner using a neural network. MoCoGAN was evaluated on unconditional video generation on the UCF-101 and Tai-Chi-HD datasets and achieved higher IS scores than the preceding TGAN and VGAN models.

**DVD-GAN** (Clark et al., 2019) uses a similar dual discriminator setup to MoCoGAN. The main difference is that DVD-GAN does not use auto-regressive prediction but instead generates all video frames in parallel. It outperformed previous methods such as TGAN-v2 and MoCoGAN on the UCF-101 dataset in terms of IS score, although DVD-GAN conditioned its generation on class labels, whereas the other approaches were unconditional.

**MoCoGAN-HD** (Tian et al., 2021) disentangles content and motion in a different way from the previous approaches. A motion generator is trained to predict a latent motion trajectory, which can then be passed as input to a fixed image generator. It outperformed previous approaches on unconditional generation in the UCF-101, Tai-Chi-HD, and Sky Time-lapse benchmarks.

**DIGAN** (Yu et al., 2022) introduces an implicit neural representation-based video GAN architecture that can efficiently represent long video sequences. It follows a similar content-motion split as discussed above. The motion discriminator judges temporal dynamics based on pairs of video frames rather than the whole sequence. These improvements enable the model to generate longer video sequences of 128 frames. DIGAN achieved state-of-the-art results on UCF-101 in terms of IS and FVD score, as well as on Sky Time-lapse and Tai-Chi-HD in terms of FVD and KVD scores.

### 7.1.2 Auto-Regressive Transformer Video Models

**VideoGPT** (Yan et al., 2021) uses a 3D VQ-VAE (Van Den Oord et al., 2017) to learn a compact video representation. An auto-regressive transformer model is then trained to predict the latent code of the next frame based on the preceding frames. While VideoGPT did not outperform the best GAN-based models at the time, namely TGAN-v2 and DVD-GAN, it achieved a respectable IS score on the UCF-101 benchmark considering its simple architecture.

**NÜWA** (Wu et al., 2022a) also uses a 3D VQ-VAE with an auto-regressive transformer generator. It is pre-trained on a variety of data sets that enable it to perform various generation and editing tasks in the video and image domains. Its text-conditioned video generation capability was evaluated on the MSR-VTT data set.

**TATS** (Ge et al., 2022) introduces several improvements that address the issue of quality degradation that auto-regressive transformer models face when generating long video sequences. It beat previous methods on almost all metrics for UFC-101 (unconditional and class-conditioned), Tai-Chi-HD, and Sky Time-lapse. Only DIGAN maintained a higher FVD score on the Sky Time-lapse benchmark.

**CogVideo** (Hong et al., 2022) is a text-conditioned transformer model. It is based on the pre-trained text-to-image model CogView2 (Ding et al., 2022), which is expanded with spatio-temporal attention layers. The GPT-like transformer generates key frames in a latent VQ-VAE space and a second upsampling model interpolates them to a higher framerate. The model was trained on an internal data set of 5.4 mio. annotated videos with a resolution of $160 \times 160$. It was then evaluated on the UCF-101 data set in a zero-shot setting by using the 101 class labels as text prompts. It beats most other models in terms of FVD and IS score except for TATS.

### 7.1.3 Diffusion Models

Producing realistic videos based on only a text prompt is one of the most challenging tasks for video diffusion models. A key problem lies in the relative lack of suitable training data. Publicly available video data sets are usually unlabeled, and human-annotated labels may not even accurately describe the complex relationship between spatial and temporal information. Many authors therefore supplement training of their models with large data sets of labeled images or build on top of a pre-trained text-to-image model. The first video diffusion models (Ho et al., 2022c) had very high computational demands paired with relatively low visual fidelity. Both aspects have significantly been improved through architectural advancements, such as moving the denoising process to the latent space of a variational auto-encoder (He et al., 2022b; Zhou et al., 2022; Chen et al., 2023a; 2024; Blattmann et al., 2023a; Zhang et al., 2023a) and using upsampling techniques such as CDMs (Ho et al. 2022a; Wang et al. 2023c, see section 4.3).

Ho et al. (2022c) present an early diffusion-based video generation model called **VDM**. It builds on the 3D UNet architecture proposed by Çiçek et al. (2016), extending it by factorized spatio-temporal attention blocks. This produces videos that are 16 frames long and $64 \times 64$ pixels large. These low-resolution videos can then be extended to $128 \times 128$ pixels and 64 frames using a larger upsampling model. The models are trained on a relatively large data set of labeled videos as well as single frames from those videos, which enables text-guided video generation at time of inference. However, this poses a limitation of this approach since labeled video data is relatively difficult to come by.

Singer et al.'s (2022) **Make-a-Video** address this issue by combining supervised training of their model on labeled images with unsupervised training on unlabeled videos. This allows them to access a wider and more diverse pool of training data. They also split the convolution layers in their UNet model into 2D spatial convolutions and 1D temporal convolutions, thereby alleviating some of the computational burden associated with a full 3D UNet. Finally, they train a masked spatiotemporal decoder on temporal upsampling or video prediction tasks. This enables the generation of longer videos of up to 76 frames. Make-a-Video was evaluated on the UCF-101 and MSR-VTT benchmarks where it outperformed all previous GAN and autoregressive transformer models.

Ho et al. (2022a) use a cascaded diffusion process (Ho et al. 2022b, see Figure 11) that can generate high-resolution videos in their model called **ImagenVideo**. They start with a base model that synthesizes videos with $40 \times 24$ pixels and 16 frames, and upsample it over six additional diffusion models to a final resolution of $1280 \times 768$ pixels and 128 frames. The low-resolution base model uses factorized space-time convolutions and attention. To preserve computational resources, the upsampling models only rely on convolutions. ImagenVideo is trained on a large proprietary data set of labeled videos and images in parallel, enabling it to emulate a variety of visual styles. The model also demonstrates the ability to generate animations of text, which most other models struggle with.

Zhou et al.'s (2022) **MagicVideo** adapts the Latent Diffusion Models (Rombach et al. 2022, see Figure 11) architecture for video generation tasks. In contrast to the previous models that operate in pixel space, their diffusion process takes place in a low-dimensional latent embedding space defined by a pre-trained variational auto-encoder (VAE). This significantly improves the efficiency of the video generation process. This VAE is trained on video data and can thereby reduce motion artifacts compared to VAEs used in text-to-image models. The authors use a pre-trained text-to-image model as the backbone of their video model with added causal attention blocks. The model is fine-tuned on data sets of labeled and unlabeled videos. It produces videos of 256×256 pixels and 16 frames that can be upsampled using separate spatial and temporal super-resolution models to 1024×1024 pixels and 61 frames. In addition to text-to-video generation, the authors also demonstrate video editing and image animation capabilities of their model.

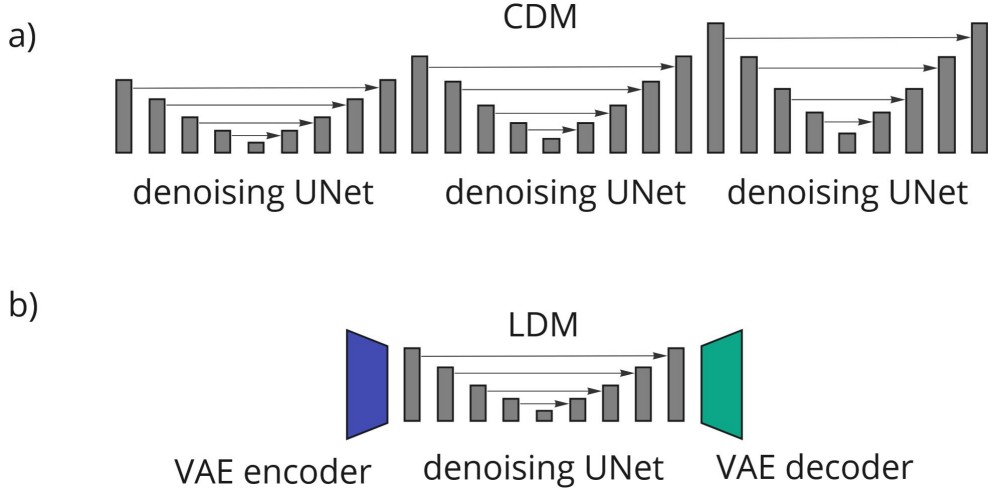

Figure 11: Architectural choices for increasing the output resolution of image diffusion models. a) Cascaded Diffusion Models (CDM) chain denoising UNets of increasing resolution to generate high-fidelity images. b) Latent Diffusion Models (LDM) use a pre-trained variational auto-encoder (VAE) to operate in lower-dimensional space, thus preserving computational resources.

Blattmann et al. (2023b) present another adaptation of the Latent Diffusion Models (Rombach et al., 2022) architecture to text-to-video generation tasks called **VideoLDM**. Similar to Zhou et al. (2022), they add temporal attention layers to a pre-trained text-to-image diffusion model and fine-tune them on labeled video data. They demonstrate that, in addition to text-to-video synthesis, their model is capable of generating long driving car video sequences in an auto-regressive manner, as well as of producing videos of personalized characters using Dreambooth (Ruiz et al., 2023).

Khachatryan et al.'s (2023) **Text2Video-Zero** completely eschews the need for video training data, instead relying only on a pre-trained text-to-image diffusion model to perform zero-shot text-to-video generation. Specifically, to ensure the appearance consistency of the objects across different frames, they equip the pretrained image diffusion model with cross-frame attention blocks such that each frame can attend to all the other frames during generation. Motion is simulated by applying a warping function to latent frames, although it has to be mentioned that the resulting movement lacks realism compared to models trained on video data. Spatio-temporal consistency is improved by masking foreground objects with a trained object detector network and smoothing the background across frames. Similar to Zhou et al. (2022), the diffusion process takes place in latent space.

Guo et al. (2023) offer a text-to-video model developed with personalized image generation in mind. Their **AnimateDiff** extends a pre-trained Stable Diffusion model with a temporal adapter module merely containing self-attention blocks. During training, only the weights in the temporal adapter modules are finetuned on video data, and the image diffusion backbone remains untouched. In this way, simple movement can be induced. The authors also demonstrate that after training, the same temporal adapter module can be

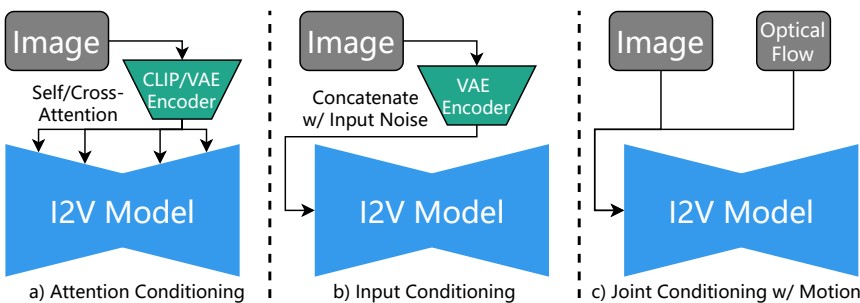

Figure 12: Image conditioning methods for image-to-video generation models. a) input images can be conditioned in the attention layers of the video generation models. b) input images can be formed as the extra input channels to the diffusion models. c) input images can be jointly conditioned with other modalities, such as optical flow.

directly applied to different image diffusion backbones, making their approach compatible with personalized image generation techniques such as Dreambooth (Ruiz et al., 2023) and LoRA (Hu et al., 2021).

## 7.2 Image-Conditioned Generation

An important limitation of text-to-video models is the lack of controllability, as the video content can only be determined by an input text prompt. To mitigate this issue, recent research has been focusing on introducing additional image conditional signals to the video generation process. Image conditioning can be achieved through injecting semantic image embeddings (e.g. CLIP (Radford et al., 2021) image embeddings) (Wang et al., 2024a; Zhang et al., 2023b; Xing et al., 2023b; Chen et al., 2023a) or image VAE latents (Ren et al., 2024; Zhang et al., 2024) in attention layers (Figure 12 a)), adding extra input channels that represent the conditioning image (Figure 12 b)) (Girdhar et al., 2023; Chen et al., 2023e; Zeng et al., 2023; Blattmann et al., 2023a), joint conditioning with other modalities such as optical flows (Figure 12 c))(Chen et al., 2023d; Shi et al., 2024), etc. Image-conditioned generation also enables a wide range of applications, such as autoregressive long video generation (Zeng et al., 2023; Chen et al., 2023e; Ren et al., 2024), looping video generation (Xing et al., 2023b), generative frame interpolation (Zeng et al., 2023; Xing et al., 2023b) and visual storytelling (Zeng et al., 2023; Xing et al., 2023b).

Chen et al. (2023d) focus on the task of animating images in accordance with motion cues. Their Motion-Conditioned Diffusion Model (**MCDiff**) accepts an input image and lets the user indicate the desired motion by drawing strokes on top of it. The model then produces a short video sequence in which objects move in accordance with the motion cues. It can dissociate between foreground (e.g. actor movement) or background motion (i.e. camera movement), depending on the context. The authors use an auto-regressive approach to generate each video frame conditioned on the previous frame and predicted motion flow. For this, the input motion strokes are decomposed into smaller segments and passed to a UNet flow completion model to predict motion in the following frame. A denoising diffusion model receives this information and uses it to synthesize the next frame. The flow completion model and the denoising model are first trained separately but later fine-tuned jointly on unannotated videos.

Chen et al.'s (2023e) **SEINE** proposes to train an image-conditioned video generation model by concatenating the VAE latent of the image along the channel dimension of the input noise and adding an extra mask channel indicating which frame needs to be predicted. This enables flexible image conditioning such that the model can generate videos providing any given frames as conditional signals. SEINE is initialized from the text-to-video model LaVie (Wang et al., 2023c) and trained on WebVid-10M (Bain et al., 2021) along with internal private data. During inference, the model is able to perform autoregressive long video generation (by reusing the last frame of a previous video clip as the first frame to predict the next video), generating transitions between different scenes (by using two frames from different scenes as the conditioning first frame and last frame and generate the intermediate frames) and image animation (by conditioning the video generation process on the input first frame).

# 8  Video Completion & Long Video Generation

As mentioned in Section 5, video diffusion models typically generate a fixed number of frames at a time. Additionally, the number of frames produced in one denoising process is usually limited to a small quantity, such as 8 or 16 frames. As a result, this limitation leads to the generation of very short videos with low frame rates (e.g. 2 seconds, 8 fps). In order to circumvent this limitation, auto-regressive extension and temporal upsampling methods have been proposed (see Section 5.2) to enhance the duration and frame rate of the generated videos. Models adopting these methods often adjust and combine them in unique ways that benefit computational speed or consistency. A common problem with these approaches is that they tend to generate videos that suffer from repetitive content. Some models have therefore explored ways to generate videos with changing scenes by varying the text prompts over time.

## 8.1  Temporal Upsampling & Video Prediction

Yin et al.'s (2023) **NUWA-XL** model uses an iterative hierarchical approach to generate long video sequences of several minutes. It first generates evenly spaced key frames from separate text prompts that form a rough outline of the video. The frames in-between are then filled in with a local diffusion model conditioned on two key frames. This process is applied iteratively to increase the temporal resolution with each pass. Since this can be parallelized, the model achieves much faster computation times than auto-regressive approaches for long video generation. The authors train the model on a new training data set consisting of annotated Flintstones cartoons. Simple temporal convolution and attention blocks are inserted into the pre-trained text-to-image model to learn temporal dynamics.

He et al. (2022b) tackle the task of generating long videos with over 1,000 frames with their Long Video Diffusion Model (**LVDM**). It combines auto-regressive and hierarchical approaches for first generating long sequences of key frames and then filling in missing frames. In order to reduce quality degradation induced by auto-regressive sampling, the authors use classifier-free guidance and conditional latent perturbation which conditions the denoising process on noisy latents of reference frames. The model utilizes a dedicated video encoder and combines 2D spatial with 1D temporal self-attention. It can be used for unconditional video generation or text-to-video tasks.

Harvey et al. (2022) similarly explore methods for generating long video sequences with video models that have a fixed number of output frames. Their Flexible Diffusion Model (**FDM**) accepts an arbitrary number of conditioning frames to synthesize new frames, thereby allowing it to either extend the video in an auto-regressive manner or to use a hierarchical approach (similar to NUWA-XL, Yin et al. 2023). The authors explore variations of these sampling techniques and suggest an automated optimization routine that finds the best one for a given training data set.

Lu et al. (2023b) propose Video Diffusion Transformer (**VDT**), a diffusion-based video model that uses a vision transformer architecture (Peebles & Xie, 2022). The reported advantages of this type of architecture over the commonly used UNet include the ability to capture long-range temporal dynamics, to accept conditioning inputs of varying lengths, and the scalability of the model. VDT was trained on more narrow data sets of unlabeled videos and accomplished tasks such as video prediction, temporal interpolation, and image animation in those restricted domains.

## 8.2  Alternative Approaches

Wang et al.'s (2023) **Gen-L-Video** generates long video sequences by denoising overlapping shorter video segments in parallel. A video diffusion model predicts the denoised latent in each video segment individually. The noise prediction for a given frame is than aggregated through interpolation across all segments in which it appears. This leads to greater coherence across the long video sequence. The authors apply this new method to existing frameworks in the text-to-video (LVDM, He et al. 2022b), tuning-free video-to-video (Pix2Video, Ceylan et al. 2023), and one-shot tuning video-to-video (Tune-A-Video, Wu et al. 2022b) domains.

Zhu et al. (2023) follow a unique approach for generating long video sequences in their **MovieFactory** model. Rather than extending a single video clip, they generate a movie-like sequence of separate related

clips from a single text prompt. ChatGPT is used to turn the brief text prompt into ten detailed scene descriptions. Each scene description is then passed as a prompt to the video diffusion model to generate a segment of the video sequence. Finally, audio clips matching each video scene are retrieved from a sound database. The pre-trained text-to-image model (Stable Diffusion 2.0) is first expanded by additional ResNet and attention blocks that are trained in order to produce wide-screen images. In a second training step, 1D temporal convolution and attention blocks are added to learn temporal dynamics.

Sun et al.'s (2023) **GLOBER** is a model for generating videos of arbitrary length that does not rely on auto-regressive or hierarchical approaches. Instead, it first uses a video KL-VAE auto-encoder to extract global 2D features from key frames. It then provides these global features along with arbitrary frame indices to a UNET diffusion model that can directly generate frames at those positions. To ensure the temporal coherence and realism of the generated frames, a novel adversarial loss is introduced. During training, an adversarial discriminator model receives pairs of video frames at random positions along with their indices and has to predict whether the frames both originated from the input video, or whether one or both were generated by the diffusion model. To enable inference, a generator model based on the Diffusion Transformer architecture (Peebles & Xie, 2022) is trained to produce global features that mimic those of the video encoder given text prompts. GLOBER surpasses several competing models in terms of FVD score, but its main advantage is a much faster computation time compared to auto-regressive methods.

Luo et al. (2023) improve the temporal coherence in their **VideoFusion** model by decomposing the noise added during the forward diffusion process. A base noise component is shared across all frames and characterizes the content of the entire video. Meanwhile, a residual component is specific to each frame and is partially related to the motion of objects. This approach saves computational resources since a smaller residual generator denoising model can be used to estimate the residual noise for each frame, whereas the base noise has to be estimated only once for the entire video using a pre-trained image model. The pre-trained base generator is fine-tuned jointly with the residual generator.

Hu et al.'s (2023a) **GAIA-1** is a hybrid transformer-diffusion model that can generate driving car videos conditioned on images, text, or action tokens that represent speed and movement trajectories. During training, it first uses a VQ-GAN to transform input video frames into discrete tokens. An auto-regressive transformer world model is used to predict the next token in the sequence based on all preceding tokens using causal masking. A diffusion-based video decoder then translates the tokens back to pixel space by denoising random noise patterns conditioned on the generated token sequence. The decoder is trained to enable flexible applications such as auto-regressive video generation and frame interpolation.

## 9 Audio-conditioned Synthesis

Multimodal synthesis might be the most challenging task for video diffusion models. A key problem lies in how associations between different modalities can be learned. Similar to how CLIP models (Radford et al., 2021) encode text and images in a shared embedding space, many models learn a shared semantic space for audio, text, and / or video through techniques such as contrastive learning (Chen et al., 2020).

### 9.1 Audio-conditioned Generation & Editing

Lee et al.'s (2023a) **Soundini** model enables local editing of scenic videos based on sound clips. A binary mask can be specified to indicate a video region that is intended to be made visually consistent with the auditory contents of the sound clip. To this end, a sliding window selection of the sound clip's mel spectrogram is encoded into a shared audio-image semantic space. During training, two loss functions are minimized to condition the denoising process on the embedded sound clips: The cosine similarity between the encoded audio clip and the image latent influences the generated video content, whereas the cosine similarity between the image and audio gradients is responsible for synchronizing the video with the audio signal. In contrast to other models, Soundini does not extend its denoising UNet to the video domain, only generating single frames in isolation. To improve temporal consistency, bidirectional optical flow guidance is used to warp neighboring frames towards each other.

Table 4: Video generation benchmarks.

| Model | Resolution | Zero-Shot | Conditioning | Training | UCF-101 | | | MSR-VTT | | | Tai-Chi-HD | | Sky Time-lapse | |
|---|---|---|---|---|---|---|---|---|---|---|---|---|---|---|
| | | | | | FID↓ | FVD↓ | IS↑ | CLIP-Sim↑ | FID↓ | FVD↓ | FVD↓ | KVD↓ | FVD↓ | KVD↓ |
| **GAN Models** | | | | | | | | | | | | | | |
| MoCoGAN (2018) | 64×64 | No | - | - | 26998 | - | 12.42 | - | - | - | - | - | - | - |
| TGAN-v2 (2020) | 64×64 | No | - | - | 3431 | - | 26.6 | - | - | - | - | - | - | - |
| TGAN-v2 (2020) | 128×128 | No | - | - | 3497 | - | 28.87 | - | - | - | - | - | - | - |
| TGAN-F (2020) | 64×64 | No | - | - | 8942 | - | 13.62 | - | - | - | - | - | - | - |
| TGAN-F (2020) | 128×128 | No | - | - | 7817 | - | 22.91 | - | - | - | - | - | - | - |
| DVD-GAN (2019) | 128×128 | No | Class | - | - | - | 32.97 | - | - | - | - | - | - | - |
| MoCoGAN-HD (2021) | 256×256 | No | - | UCF-101: test split incl. | - | 700 | 34 | - | - | - | 144.7 | 25.4 | 183.6 | 13.9 |
| DIGAN (2022) | 128×128 | No | - | - | - | 655 | 29.7 | - | - | - | 128.1 | 20.6 | 114.6 | 6.8 |
| DIGAN (2022) | 128×128 | No | - | UCF-101: test split incl. | - | 577 | 32.7 | - | - | - | 128.1 | 20.6 | 114.6 | 6.8 |
| **Transformer Models** | | | | | | | | | | | | | | |
| VideoGPT (2021) | 128×128 | No | - | - | - | - | 24.69 | - | - | - | - | - | - | - |
| NUWA (2022a) | 128×128 | No | - | VATEX | - | - | - | 0.2439 | 47.7 | - | - | - | - | - |
| TATS-base (2022) | 128×128 | No | - | - | - | 420 | 57.6 | - | - | - | 94.6 | 8.8 | 132.6 | 5.7 |
| TATS-base (2022) | 128×128 | No | Class | - | - | 332 | 79.3 | - | - | - | - | - | - | - |
| CogVideo (Chinese) (2022) | 480×480 | Yes | Text | internal data | 185 | 751.3 | 23.6 | 0.2614 | 24.8 | - | - | - | - | - |
| CogVideo (English) (2022) | 480×480 | Yes | Text | internal data | 179 | 701.6 | 25.3 | 0.2631 | 23.6 | - | - | - | - | - |
| CogVideo (English) (2022) | 160×160 | Yes | Text | internal data | - | 626 | 50.5 | - | 49 | 1294 | - | - | - | - |
| **Diffusion Models** | | | | | | | | | | | | | | |
| VDM (2022c) | 64×64 | No | - | - | 295 | - | 57 | - | - | - | - | - | - | - |
| Make-a-Video (2022) | 256×256 | Yes | Text | WebVid10M, HD-VILA-10M | - | 367.2 | 33 | **0.3049** | **13.2** | - | - | - | - | - |
| Make-a-Video (2022) | 256×256 | No | Text | WebVid10M, HD-VILA-10M | - | **81.3** | **82.6** | - | - | - | - | - | - | - |
| MagicVideo (2022) | 256×256 | Yes | Text | WebVid10M, HD-VILA-100M | **145** | 665 | - | - | 36.5 | **998** | - | - | - | - |
| LVDM (2022b) | 256×256 | No | - | UCF-101: test split incl. | - | 372 | - | - | - | - | 99 | 15.3 | 95.2 | **3.9** |
| VideoLDM (SD 1.4) (2023b) | 1280×2048 | Yes | Text | WebVid-10M | - | 656.5 | 29.5 | 0.2848 | - | - | - | - | - | - |
| VideoLDM (SD 2.1) (2023b) | 1280×2048 | Yes | Text | WebVid-10M | - | 550.6 | 33.5 | 0.2929 | - | - | - | - | - | - |
| PYoCo (2023) | 1024×1024 | Yes | Text | internal data | - | 355.19 | 47.76 | - | 22.14 | - | - | - | - | - |
| Make-Your-Video (2023a) | 256×256 | Yes | Text + Depth | WebVid-10M | - | 330.5 | - | - | - | - | - | - | - | - |
| VDT (2023b) | 64×64 | No | - | - | - | 225.7 | - | - | - | - | - | - | - | - |
| VideoFusion (2023) | 128×128 | No | - | - | - | 220 | 72.2 | - | - | - | **56.4** | **6.9** | 47 | 5.3 |
| VideoFusion (2023) | 128×128 | No | Text | - | - | 173 | 80 | - | - | - | - | - | - | - |
| GLOBER (2023) | 128×128 | No | - | - | - | 239.5 | - | - | - | - | 124.2 | - | - | - |
| GLOBER (2023) | 128×128 | No | Text | - | - | 151.5 | - | - | - | - | - | - | - | - |
| GLOBER (2023) | 256×256 | No | - | - | - | 252.7 | - | - | - | - | - | - | 78.1 | - |
| GLOBER (2023) | 256×256 | No | Text | - | - | 168.9 | - | - | - | - | - | - | - | - |
| LaVie (2023c) | 512×320 | Yes | Text | Vimeo25M | - | 526.3 | - | 0.2949 | - | - | - | - | - | - |

Lee et al. (2023b) generate scenic videos from text prompts and audio clips with their Audio-Aligned Diffusion Framework (**AADiff**). An audio clip is used to identify a target token from provided text tokens, based on the highest similarity of the audio clip embedding with one of the text token embeddings. For instance, a crackling sound might select the word "burning". While generating video frames, the influence of the selected target token on the output frame is modulated through attention map control (similar to Prompt-to-Prompt, Hertz et al. 2022) in proportion to the sound magnitude. This leads to changes in relevant video elements that are synchronized with the sound clip. The authors also demonstrate that their model can be used to animate a single image and that several sound clips can be inserted in parallel. The model uses a pre-trained text-to-image model to generate each video frame without additional fine-tuning on videos or explicit modeling of temporal dynamics.

Liu et al.'s (2023d) **Generative Disco** provides an interactive interface to support the creation of music visualizations. They are implemented as visual transitions between image pairs created with a diffusion model from user-specified text prompts. The interval in-between the two images is filled according to the beat of the music, using a form of interpolation that employs design patterns to cause shifts in color, subject, or style, or set a transient video focus on subjects. A large language model can further assist the user with choosing suitable prompts. While the model is restricted to simple image transitions and is therefore not able to produce realistic movement, it highlights the creative potential of video diffusion models for music visualization.

Tang et al. (2023) present a model called **Composable Diffusion** that can generate any combination of output modalities based on any combination of input modalities. This includes text, images, videos, and sound. Encoders for the different modalities are aligned in a shared embedding space through contrastive learning. The diffusion process can then be flexibly conditioned on any combination of input modalities by linearly interpolating between their embeddings. A separate denoising diffusion model is trained for each of the output modalities and information between the modality-specific models is shared through cross-attention blocks. The video model uses simple temporal attention as well as the temporal shift method from An et al. (2023) to ensure consistency between frames.

## 9.2 Talking Head Generation

Stypułkowski et al. (2023) have developed the first diffusion model for generating videos of talking heads. Their model **Diffused Heads** takes a reference image of the intended speaker as well as a speech audio clip as input. The audio clip is divided into short chunks that are individually embedded through a pre-trained audio encoder. During inference, the reference image as well as the last two generated video frames are concatenated with the noisy version of the current video frame and passed through a 2D UNet. Additionally, the denoising process is conditioned on a sliding window selection of the audio embeddings. The generated talking faces move their lips in sync with the audio and display realistic facial expressions.

Zhua et al. (2023) follow a similar approach, but instead of using a reference image, their model accepts a reference video that is transformed to align with the desired audio clip. Face landmarks are first extracted from the video, and then encoded into eye blink embeddings and mouth movement embeddings. The mouth movements are aligned with the audio clip using contrastive learning. Head positions and eye blinks are encoded with a VAE, concatenated together with the synchronized mouth movement embeddings, and passed as conditioning information to the denoising UNet.

Casademunt et al. (2023) focus on the unique task of laughing head generation. Similar to Diffused Heads (Stypułkowski et al., 2023), the model takes a reference image and an audio clip of laughter to generate a matching video sequence. The model combines 2D spatial convolutions and attention blocks with 1D temporal convolutions and attention. This saves computational resources over a fully 3D architecture and allows it to process 16 video frames in parallel. Longer videos can be generated in an auto-regressive manner. The authors demonstrate the importance of using a specialized audio-encoder for embedding the laughter clips in order to generate realistic results.

# 10 Video Editing

Editing can mean a potentially wide range of operations such as adjusting the lighting, style, or background, changing, replacing, re-arranging, or removing objects or persons, modifying movements or entire actions, and more. To avoid having to make cumbersome specifications for possibly a large number of video frames, a convenient interface is required. To achieve this, most approaches rely on textual prompts that offer a flexible way to specify desired edit operations at a convenient level of abstraction and generality. However, completely unconstrained edit requests may be in conflict with desirable temporal properties of a video, leading to a major challenge of how to balance temporal consistency and editability (see Section 5.3). To this end, many authors have experimented with conditioning the denoising process based on preprocessed features of the input video. One-shot tuning methods first fine-tune their parameters on the ground truth video. This ensures that the video content and structure can be reconstructed with good quality. On the other hand, tuning-free methods are not fine-tuned on the ground truth video, which makes the editing computationally more efficient.

## 10.1 One-Shot Tuning Methods

Molad et al. (2023) present a diffusion video editing model called **Dreamix** based on the ImagenVideo (Ho et al., 2022a) architecture. It first downsamples an input video, adds Gaussian noise to the low resolution version, then applies a denoising process conditioned on a text prompt. The model is finetuned on each input video and follows the joint training objective of preserving the appearance of both the entire video and individual frames. The authors demonstrate that the model can edit the appearance of objects as well as their actions. It is also able to take either a single input image or a collection of images depicting the same object and animate it. Like ImagenVideo (Ho et al., 2022a), Dreamix operates in pixel space rather than latent space. Together with the need to finetune the model on each video, this makes it computationally expensive.

Wu et al. (2022b) base their **Tune-A-Video** on a pre-trained text-to-image diffusion model. Rather than fine-tuning the entire model on video data, only the projection matrices in the attention layers are trained on a given input video. The spatial self-attention layer is replaced with a spatio-temporal layer attending to

previous video frames, while a new 1D temporal attention layer is also added. The structure of the original frames is roughly preserved by using latents obtained with DDIM inversion as the input for the generation process. The advantages of this approach are that fine-tuning the model on individual videos is relatively quick and that extensions developed for text-to-image tasks such as ControlNet (Zhang & Agrawala, 2023) or Dreambooth (Ruiz et al., 2023) can be utilized. Several models have subsequently built upon the Tune-A-Video approach and improved it in different ways:

Qi et al. (2023) employ an attention blending method inspired by Prompt-to-Prompt (Hertz et al., 2022) in their **FateZero** model. They first obtain a synthetic text description of the middle frame from the original video through BLIP (Li et al., 2022) that can be edited by the user. While generating a new image from the latent obtained through DDIM inversion, they blend self- and cross-attention masks of unedited words with the original ones obtained during the inversion phase. In addition to this, they employ a masking operation that limits the edits to regions affected by the edited words in the prompt. This method improves the consistency of generated videos while allowing for greater editability compared to Tune-A-Video.

Liu et al. (2023b) also base their **Video-P2P** model on Tune-A-Video and similar to FateZero, they incorporate an attention-tuning method inspired by Prompt-to-Prompt. Additionally, they augment the DDIM inversion of the original video by using Null-text inversion (Mokady et al., 2023), thereby improving its reconstruction ability.

## 10.2    Depth-conditioned Editing

Ceylan et al.'s (2023) **Pix2Video** continues the trend of using a pre-trained text-to-image model as the backbone for video editing tasks. In contrast to the previous approaches, it however eliminates the need for fine-tuning the model on each individual video. In order to preserve the coarse spatial structure of the input, the authors use DDIM inversion and condition the denoising process on depth maps extracted from the original video. Temporal consistency is ensured by injecting latent features from previous frames into self-attention blocks in the decoder portion of the UNet. The projection matrices from the stock text-to-image model are not altered. Despite using a comparatively lightweight architecture, the authors demonstrate good editability and consistency in their results.

Esser et al.'s (2023) **Runway Gen-1** enables video style editing while preserving the content and structure of the original video. This is achieved on the one hand by conditioning the diffusion process on CLIP embeddings extracted from a reference video frame (in addition to the editing text prompt), and on the other hand by concatenating extracted depth estimates to the latent video input. The model uses 2D spatial and 1D temporal convolutions as well as 2D + 1D attention blocks. It is trained on video and image data in parallel. Predictions of both modes are combined in a way inspired by classifier-free guidance (Ho & Salimans, 2022), allowing for fine-grained control over the tradeoff between temporal consistency and editability. The successor model Runway Gen-2 (unpublished) also adds image-to-video and text-to-video capabilities.

Xing et al. (2023a) extend a pre-trained text-to-image model conditioned on depth maps to video editing tasks in their **Make-Your-Video** model, similar to Pix2Video (Ceylan et al., 2023). They add 2D spatial convolution and 1D temporal convolution layers, as well as cross-frame attention layers to their UNet. A causal attention mask limits the number of reference frames to the four immediately preceding ones, as the authors note that this offers the best trade-off between image quality and coherence. The temporal modules are trained on a large unlabeled video data set (WebVid-10M, Bain et al. 2021).

## 10.3    Pose-conditioned Editing

Ma et al.'s (2023) **Follow Your Pose** conditions the denoising process in Tune-A-Video on pose features extracted from an input video. The pose features are encoded and downsampled using convolutional layers and passed to the denoising UNet through residual connections. The pose encoder is trained on image data, whereas the spatio-temporal attention layers (same as in Tune-A-Video) are trained on video data. The model generates output that is less bound by the source video while retaining relatively natural movement of subjects.

Zhao et al.'s (2023) **Make-A-Protagonist** combines several expert models to perform subject replacement and style editing tasks. Their pipeline is able to detect and isolate the main subject (i.e. the "protagonist") of a video through a combination of Blip-2 (Li et al., 2023b) interrogation, Grounding DINO (Liu et al., 2023c) object detection, Segment Anything (Kirillov et al., 2023) object segmentation, and XMem (Cheng & Schwing, 2022) mask tracking across the video. The subject can then be replaced with that from a reference image through Stable Diffusion inpainting with ControlNet depth map guidance. Additionally, the background can be changed based on a text prompt. The pre-trained Stable Diffusion UNet model is extended by cross-frame attention and fine-tuned on frames from the input video.

## 10.4 Leveraging Pre-trained Video Generation Model for Video Editing

Instead of adapting a pre-trained image generation model for video editing, Bai et al.'s (2024) **UniEdit** investigates the approach of leveraging a pre-trained text-to-video generation model for zero-shot video editing. Specifically, they propose to use the LaVie (Wang et al., 2023c) T2V model and employ feature injection mechanisms to condition the T2V generation process on the input video. This is achieved by introducing the auxiliary reconstruction branch and motion-reference branch during video denoising. The video features from these auxiliary branches are extracted and injected into the spatial and temporal self-attention layers of the main editing path to ensure the output video contains the same spatial structure and motion as the source video.

A concurrent approach of UniEdit is Ku et al.'s (2024) **AnyV2V**, which employs pre-trained image-to-video (I2V) generation models for zero-shot video editing tasks. AnyV2V breaks video editing into two stages. In the first stage, an image editing method is used to modify the first frame of the video into an edited frame. In the second stage, the edited frame and the DDIM inverted latent of the source video are passed into the I2V generation model to render the edited video. AnyV2V also adopts feature injection mechanisms similar to PnP (Tumanyan et al., 2023) to preserve the structure and motion of the source video. Because of the proposed two-stage editing strategy, AnyV2V is compatible with any off-the-shelf image editing models and can be employed in a broad spectrum of video editing tasks, such as prompt-based video editing, reference-based style transfer, identity manipulation and subject-driven video editing. The framework also supports different I2V models, such as I2VGen-XL (Zhang et al., 2023b), ConsistI2V (Ren et al., 2024) and SEINE (Chen et al., 2023e).

## 10.5 Multi-conditional Editing

Zhang et al.'s (2023c) **ControlVideo** model extends ControlNet (Zhang & Agrawala, 2023) to video generation tasks. ControlNet encodes preprocessed image features using an auto-encoder and passes them through a fine-tuned copy of the first half of the Stable Diffusion UNet. The resulting latents at each layer are then concatenated with the corresponding latents from the original Stable Diffusion model during the decoder portion of the UNet to control the structure of the generated images. In order to improve the spatio-temporal coherence between video frames, ControlVideo adds full cross-frame attention to the self-attention blocks of the denoising UNet. Furthermore, it mitigates flickering of small details by interpolating between alternating frames. Longer videos can be synthesized by first generating a sequence of key frames and then generating the missing frames in several batches conditioned on two key frames each. In contrast to other video-to-video models that rely on a specific kind of preprocessed feature, ControlVideo is compatible with all ControlNet models, such as Canny or OpenPose. The pre-trained Stable Diffusion and ControlNet models also do not require any fine-tuning.

## 10.6 Other Approaches

Wang et al. (2023b) also adapt a pre-trained text-to-image model to video editing tasks without fine-tuning. Similar to Tune-A-Video and Pix2Video, their **vid2vid-zero** model replaces self-attention blocks with cross-frame attention without changing the transformation matrices. While the cross-frame attention in those previous models is limited to the first and immediately preceding frame, Wang et al. extend attention to the entire video sequence. Vid2vid-zero is not conditioned on structural depth maps, instead using a traditional

DDIM inversion approach. To achieve better alignment between the input video and user-provided prompt, it optimizes the null-text embedding used for classifier-free guidance.

Huang et al. (2023) present **Style-A-Video**, a model aimed at editing the style of a video based on a text prompt while preserving its content. It utilizes a form of classifier-free guidance that balances three separate guidance conditions: CLIP embeddings of the original frame preserve semantic information, CLIP embeddings of the text prompt introduce stylistic changes, while CLIP embeddings of thresholded affinity matrices from self-attention layers in the denoising UNet encode the spatial structure of the image. Flickering is reduced through a flow-based regularization network. The model operates on each individual frame without any form of cross-frame attention or fine-tuning of the text-to-image backbone. This makes it one of the lightest models in this comparison.

Yang et al. (2023b) also use ControlNet for spatial guidance in their **Rerender A Video** model. Similar to previous models, sparse causal cross-frame attention blocks are used to attend to an anchor frame and the immediately preceding frame during each denoising step. During early denoising steps, frame latents are additionally interpolated with those from the the anchor frame for rough shape guidance. Furthermore, the anchor frame and previous frame are warped in pixel space to align with the current frame, encoded, and then interpolated in latent space. To reduce artifacts associated with repeated encoding, the authors estimate the encoding loss and shift the encoded latent along the negative gradient of the loss function to counteract the degradation. A form of color correction is finally applied to ensure color coherence across frames. This pipeline is used to generate key frames that are then filled in using patch-based propagation. The model produces videos that look fairly consistent when showing slow moving scenes but struggles with faster movements due to the various interpolation methods used.

### 10.7 Video Restoration

Liu et al. (2023a) present **ColorDiffuser**, a model specialized on colorization of grayscale video footage. It utilizes a pre-trained text-to-image model and specifically trained adapter modules to colorize short video sequences in accordance with a text prompt. Color Propagation Attention computes affinities between the current grayscale frame as Query, the reference grayscale frame as Key, and the (noisy) colorized reference frame latent as Value. The resulting frame is concatenated with the current grayscale frame and fed into a Coordinator Module that follows the same architecture as the Stable Diffusion UNet. Feature maps from the Coordinator module are then injected into the corresponding layers of the denoising UNet to guide the diffusion process (similar to ControlNet). During inference, an alternating sampling strategy is employed, whereby the previous and following frame are in turn used as reference. In this way, color information can propagate through the video in both temporal directions. Temporal consistency and color accuracy is further improved by using a specifically trained vector-quantized variational auto-encoder (VQVAE) that decodes the entire denoised latent video sequence.

## 11 Video Diffusion Models for Intelligent Decision Making

Capable generative models are beginning to see widespread usage in control and intelligent decision-making (Yang et al., 2023a; Collaboration et al., 2024), including for downstream representation learning, world modeling, and generative data augmentation. So far, use cases have primarily focused on image-based and low-dimensional diffusion models, but we elucidate where these may naturally be extended to video.

### 11.1 Representation Learning

Representation learning (Bengio et al., 2013) is a popular way to transfer useful features learned from large-scale training to downstream tasks. These features usually take the form of a low-dimensional vector which can then be simply adapted to another task with a small number of linear layers. Recent work has shown that diffusion models are an effective way to do so, particularly for image and video-based tasks. A large family of methods has considered extracting representations from text-to-image diffusion models like Stable Diffusion (Rombach et al., 2022). For example, Yang & Wang (2023); Gupta et al. (2024) propose to extract representations from the diffusion model from intermediate layers of the network to be used for classification

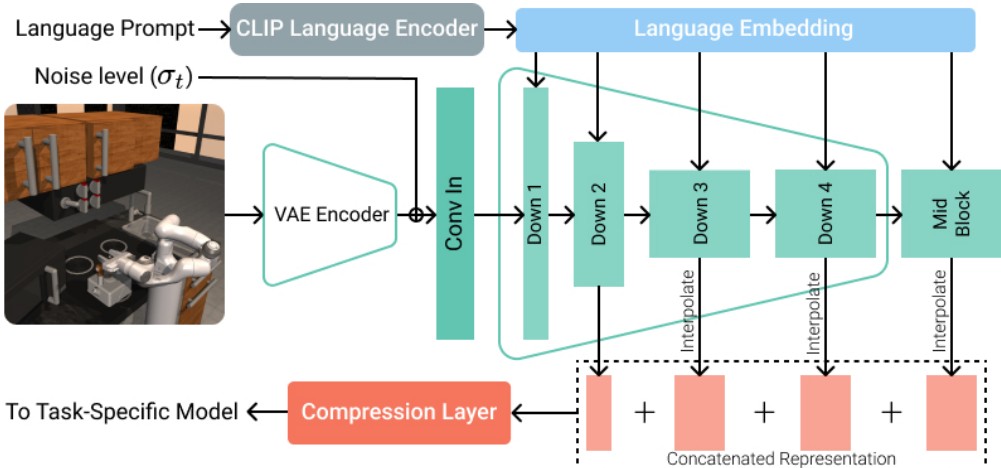

Figure 13: Depiction of how vision-language representations can be extracted from pre-trained diffusion UNets. Given an image-text prompt, one may encode and noise the image and feed it into the UNet together with the language prompt. Features may then be aggregated from multiple levels of the downsampling process. Similar techniques may be extended to video diffusion UNets. Reproduced with permission from Gupta et al. (2024).

or robotic control tasks. This process is depicted in Figure 13, where a real image is partially noised to a low noise level to be compatible with the denoising network and then the representation is extracted by aggregating intermediate outputs from the middle of the UNet. In a similar vein, Tian et al. (2023); Wang et al. (2023) extract segmentation masks for computer vision based on intermediate attention maps from the UNet. We would expect diffusion models to be suited to this task since they have already internalized the concept of objects. These vision-language representations often significantly outperform related methods such as CLIP. Due to the similarity in architecture for image and video UNets, these methods could readily be adapted to the video domain.

On the other hand, Sariyildiz et al. (2023) pre-train a visual representation learner directly with synthetic diffusion data targeted to ImageNet classification labels. Another way pre-trained diffusion models can be used for downstream classification tasks is through likelihood-based methods. Diffusion Classifier (Li et al., 2023a) exploits the fact that diffusion models can act as conditional density models, and classify images by adding noise to them and then selecting the class label that best predicts the added noise.

## 11.2 World Models

An exciting application of more realistic video diffusion models is the ability to accurately simulate the real world. As posited by LeCun (2022), learning an accurate world model is a crucial step in the path towards autonomous intelligence, enabling an agent to robustly plan and reason about the outcome of their actions. Diffusion models have already been used as trajectory world models (Janner et al., 2022; Ajay et al., 2023) in receding horizon control style setups for low-dimensional environments. In these settings, trajectories of any arbitrary quality can be biased towards high return through classifier-guided or classifier-free guidance.

Further advances in video world modelling (Yang et al., 2024; Wang et al., 2024b; Hu et al., 2023a) could lead to similar techniques being scaled towards real-world settings. A notable example of this is GENIE (Bruce et al., 2024), a video world model (albeit not diffusion-based) trained from YouTube videos that learns to plan under latent actions. Crucially, this enables agents to be trained from synthetic environments based on the vast amounts of unlabeled video on the internet. The remaining challenges with current methods include improving the frame-by-frame consistency of generated trajectories as control policies often are very sensitive to the quality of observations, and speed of generation so that such models are useable in real-time.

### 11.3 Synthetic Training Data

Finally, as we begin to exhaust the available supply of real labeled images and video, synthetic generative data has emerged as a powerful method to augment existing training datasets for downstream tasks. In supervised learning, diffusion models have been used to generate additional class-conditional data for classification (He et al., 2022a; Azizi et al., 2023) resulting in significant boosts in performance. This enables the distillation of internet-scale knowledge into these models. With more realistic video generation, we could similarly generate data for video classification or captioning tasks.

In control, there is often a lack of readily available robotics data, and as such diffusion models are a particularly powerful method to generate policy training data for reinforcement learning agents. This could be done by simply naively upsampling existing datasets (Lu et al., 2023a) or in a guided fashion (Jackson et al., 2024) which generates training data that is on-policy with the current agent being optimized. These methods vastly improve the sample efficiency of trained agents. In the visual setting, ROSIE (Yu et al., 2023) and GenAug (Chen et al., 2023f) have considered using image diffusion models to synthesize datapoints with novel backgrounds and items in order to boost the generalization performance of learned policies. Video diffusion models represent a significant improvement to single-timestep data augmentation and would enable an agent to fully simulate the outcome of a long sequence of actions.

## 12 Ethical Considerations for Video Diffusion Models

Generative AI and AI-generated content (AIGC) have transformative potential in the realm of video diffusion models, but they also pose significant risks when misused. The ability to fabricate highly realistic videos could lead to an alarming rise in disinformation and deepfakes, where fabricated media is crafted to depict individuals saying or doing things they never actually did. Such synthetic content can manipulate public perception, distort historical events, or even tarnish reputations through malicious impersonation.

While there has been extensive research related to responsible and trustworthy AI in the field of text generation (e.g. LLaMA Guard (Inan et al., 2023) for the LLaMA (Touvron et al., 2023) language model series) and image generation (e.g. Safety checker in Stable Diffusion (Rombach et al., 2022)), the field of video generation has not yet reached the same level of scrutiny and rigor in terms of safeguarding against misuse. Current explorations include AnimateDiff (Guo et al., 2023), where a safety checker similar to that in Stable Diffusion (Rombach et al., 2022) is applied to filter NSFW contents. Wang & Yang (2024) collected VidProM, a million-scale dataset containing real prompt-video pairs generated from various Text-to-Video diffusion models. They employ the state-of-the-art NSFW model Detoxify (Wang et al., 2022) to assign NSFW probabilities to each prompt based on different aspects including toxicity, obscenity, identity attack, insult, threat, and sexual explicitness. VidProM enables potential research directions such as training specialized models to distinguish between generated and real videos, which could be beneficial for advancing safe and responsible video generation.

## 13 Outlook and Challenges

Video diffusion models have already demonstrated impressive results in a variety of use cases. However, there are still several challenges that need to be overcome before we arrive at models capable of producing longer video sequences with good temporal consistency.

One issue is the relative lack of suitable training data. While there are large data sets of labeled images that have been scraped from the internet (Sec. 6.2), the available labeled video data are much smaller in size (Sec. 6.1). Many authors have therefore reverted to training their models jointly on labeled images and unlabeled videos or fine-tuning a pre-trained text-to-image model on unlabeled video data. While this compromise allows for the learning of diverse visual concepts, it may not be ideal for capturing object-specific motion. One possible solution is to manually annotate video sequences (Yin et al., 2023), although it seems unlikely that this can be done on the scale required for training generalized video models. It is to be hoped that in the future automated annotation methods will develop that allow for the generation of accurate video descriptions (Zare & Yazdi, 2022).

An even more fundamental problem is that simple text labels are often inadequate for describing the temporally evolving content of videos. This hampers the ability of current video models to generate more complex sequences of events. For this reason, it might be beneficial to examine alternative ways to describe video content that represent different aspects more explicitly, such as the actors, their actions, the setting, camera angle, lighting, scene transitions, and so on.

A different challenge lies in the modeling of (long-term) temporal dependencies. Due to the memory limitations of current graphics cards, video models can typically only process a fixed number of video frames at a time. To generate longer video sequences, the model is extended either in an autoregressive or hierarchical fashion, but this usually introduces artifacts or leads to degraded image quality over time. Possible improvements could be made on an architectural level. Most video diffusion models build on the standard UNet architecture of text-to-image models. To capture temporal dynamics, the model is extended by introducing cross-frame convolutions and/or attention. Using full 3D spatio-temporal convolutions and attention blocks is however prohibitively expensive. Many models therefore have adopted a factorized pseudo-3D architecture, whereby a 2D spatial block is followed by a 1D temporal block. While this compromise seems necessary in the face of current hardware limitations, it stands to reason that full 3D architectures might be better able to capture complex spatio-temporal dynamics once the hardware allows it. In the meantime, other methods for reducing the computational burden of video generation will hopefully be explored. This could also enable new applications of video diffusion, such as real-time video-to-video translation.

Finally, flow matching (Lipman et al., 2022; Liu et al., 2022; Albergo et al., 2023) has recently emerged as a new family of generative models, successfully applied to large-scale image (Lipman et al., 2022; Esser et al., 2024), language (Gat et al., 2024) and video (Polyak et al., 2024) data. These models can be viewed as a more general form of generative modeling that encompasses DDPM and score-based models, offering smoother dynamics in the denoising process. Inspired by the optimal transport principle (Tong et al., 2023; Pooladian et al., 2023) and enhanced by advanced ODE sampling techniques (Shaul et al., 2024), flow matching models have demonstrated improvements in both performance and efficiency, outperforming diffusion-based models in many tasks. An intriguing future direction would be to extend these models to video generation.

## 14 Conclusion

This survey has provided an overview of the evolving field of video diffusion models, examining their potential for content creation and manipulation across various domains. We have explored the field systematically, categorizing applications by input modalities, discussing architectural choices and temporal dynamics modeling, and summarizing key developments. While progress has been made in generating, editing, and transforming video content, significant work remains to be done. As research continues, video diffusion models may influence how we create and interact with visual media, potentially opening up new applications in entertainment, education, and scientific visualization.

## Acknowledgement

This work has been partially supported by the Ministry of Culture and Science of the State of North Rhine-Westphalia, Germany, through the KI-Starter funding program.

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
