# OpenReview forum: "Video Diffusion Models: A Survey"
_TMLR — Accepted by TMLR_

### Review · Reviewer_GaEp · 2024-07-09

**Summary Of Contributions:**

this paper gives an overview over video diffusion model works, including
- a brief technical/mathematical intro to diffusion,
- discussion of the core building blocks like vision transformers, U-net, etc.,
- discussion of individual aspects/problems, like the temporal consitent modeling
- evaluations, benchmarks, metrics
- and then a survey of available work on several specific tasks:
- video generation
- video completion
- usage for decision making
- etc.
- also relation to GANs for video generation is discussed,
- and then open challenges.

**Audience:**

Yes

**Broader Impact Concerns:**

-

**Claims And Evidence:**

Yes

**Requested Changes:**

based on what i wrote above, i would advise to make the paper stronger by adding/correcting technical clarity and depth at various places, and also connecting the many individual components back to the overarching concepts/notation, to make the reader more deeply and less superficially understand the technics. in particular, i suggest to improve and correct section 3, and connect the rest of the paper more clearly to the concepts/notation of sec 3, in particular doing so for the section that introduce u-net/ViT, as well as for, e.g., sec. 11.1 (see above).

**Strengths And Weaknesses:**

overall, i found the paper interesting to read, and i think they at least partially live up to their claim of a "systematic overview" over the area
- very interesting that the authors not just go into collecting the overall video diffusion approaches, but also go across the individual dimensions of building blocks, such as architectures, temporal dynamics, etc.
- also they discuss more boradly some limitations, like the fundamental lack of controllability for such generative models
- i found sec 11 on the use for decision making very relevant, because there, in particular for safe decision making across a wide range of scenarios, the stochasticity of generative models can be a feature (diversity) instead of a bug (lack of controllability).
- there are various instructive sections, like, i quickly got the high-level idea behind LDM


so i see quite some contribution and potential in this work, but while reading i feel it does not live fully up to its potential.

one main problem that i have is the lack of technical clarity and depth at various places, and also the lack of connecting the many individual components back to the overarching concepts/notation, making many components hard to understand:
- the abstract is rather short and general, and could contain a bit more of the content
- in eq7, why are the q now additionally condition on x_0, but not in eq6? shouldnt that also already be in eq6? there is a bit of a gap here for a clean understanding.
- if there was a graphic of the process including the p and q, that would be helpful.
- around eq9: the noise terms epsilon and epsilon_theta are not properly introduced and then its hard to understand what is meant here. and how preicsely are loss 7 and loss 9 related then? one can roughly guess what is meant here, but i think with just a few more explanations and proper introduction of terms, the reader would get a clearer understanding, which is somewhat important as this is of the core of the survey.
- how come a U-net does not pass the full info between encoding and decoding simply via the residual connection? is the residual connection somehow constraint to parts of the input?
- fig6 is hard to understand. for me it is again an example of something showing some information, but for me it is hard to connect it back to the context of video diffusion. in contrast, fig7 is quite instructive.
- it is very important to understand how the U-Net/ViT building blocks are actually used in the diffusion model (denoising process i guess). this is briefly described here and there in sec4, however, it would help a lot if the actual connection to the mathematical notation would be made to see how precisely the ViT is used. is it simply the 1-step conditional of the denoising process? or the error (e.g., relating it to the epsilon_theta(x, t) from sec3)? stated differently, one gets an idea about U-net, but what i'm missing is the precise link back to the original problem of vision diffusion. what is a "denoiser"?
- in sec 11.1 i found it very interesting to read that one way to use trained diffusion models for downstream tasks is to use their latent representations. however, it remains unclear to me how this is done, and there should be more of an explanation: given a diffusion model, i.e., denoising models as presented in sec3, how specifically, from this basis, does one obtain the representations? what i don't understand is that the model maps from noise to real, right? however, a representation should map from real to noise, right? there is fig12, but for me it is again an example of showing some aspects, but leaving out the curcial aspects for the reader to make sense out of how this is specifically done for diffusion models.
- generally, i found the graphics sometimes not helpful enough for a technical paper; e.g., fig4 is comparably big in relation to the information it contains, and similar fig8.

additional improvements for writing can include:
- the structure of the paper should be made more clear, both, at the beginning in an overview, and also in the main sections, telling the reader precisely what is overviewed in that very section (e.g., sections 5 etc. seem more about surveying building blocks, while sec7 is rather about whole approaches?)

more minor things:
- in sec5, when talking about "high-dimensional" architectures for the temporal dimension, often, only the downside of high computational cost is metnioned. but i would imagine that another downside could be more variance/less bias/less robustness of the high-dimensional architecture?

in summary, overall i'm leaning accept, also given that the paper already went over one iteration of reviews and the authors seem to have added already quite some things based on that one. however, as stated above, i feel the paper does unfortunately not live up to its full potential, in particular on the level of technical clarity of explanations, and connecting individual parts more clearly to the overall diffusion concepts, making it often hard for the reader to understand what is really meant.

i'm not an expert on video diffusion, so what i cannot judge is whether all/most relevant work has been listed, or if something crucial is missing.

---

> ### Author Response · Authors · 2024-09-18
> **Improvements (part 1)**
>
> Dear reviewer,
>
> thank you very much for your time and thoughtful feedback.
>
> > the abstract is rather short and general, and could contain a bit more of the content
>
> Thanks for your feedback. We improved the abstract.
>
> > in eq7, why are the q now additionally condition on x_0, but not in eq6? shouldnt that also already be in eq6? there is a bit of a gap here for a clean understanding.
>
> We condition $q$ on $x\_0$ by introducing the term $q(x\_{t-1}|x\_t, x\_0)$ in the original Eq. (7) using Bayes' rule: $q(x\_{t-1}|x\_t, x\_0) = \frac{q(x\_t | x\_{t-1}, x\_0) \, q(x\_{t-1} | x\_0)}{q(x\_t | x\_0)}$. This ensures that each KL divergence term in Eq. (7) has analytic training objectives for the network $p\_\\theta$ at different time steps. This is a commonly-used reparameterization technique to enable efficient training for diffusion models and is proposed in DDPM [1].
>
> > if there was a graphic of the process including the p and q, that would be helpful.
>
> We visualize the forward and backward processes in the newly added Figure 3. Please note that to accommodate the diffusion models from both DDPM and score-based model perspectives, we do not explicitly denote the forward and backward processes with p and q. We visualize a diffusion process in the data space rather than in latent sapce for clarity.
>
> > around eq9: the noise terms epsilon and epsilon_theta are not properly introduced and then its hard to understand what is meant here. and how preicsely are loss 7 and loss 9 related then? one can roughly guess what is meant here, but i think with just a few more explanations and proper introduction of terms, the reader would get a clearer understanding, which is somewhat important as this is of the core of the survey.
>
> Thank you for your comment. We have added more explanations about the random noise $\\epsilon$ used to corrupt samples during training and the denoising network $\\epsilon_\\theta$. In the original Eq. (7), we show all the loss terms in KL divergence and log-likelihood. Specifically, we demonstrate that the KL divergence loss has a tractable objective in Eq. (8) thanks to Bayes' rule. The full expansion of the KL divergence terms between two Gaussians at different time steps involves coefficients shown in Eq. (8), referred to as the variational bound training objective. Empirically, diffusion-based models may benefit from dropping those coefficients and simply adopting a regression loss to predict how much noise is injected into the corrupted samples, as shown in Eq. (9).
>
> > how come a U-net does not pass the full info between encoding and decoding simply via the residual connection? is the residual connection somehow constraint to parts of the input?
>
>  Thank you for your comment, but we are a bit uncertain about which part of the paper this comment is referring to. If it is related to Section 4.1, we would like to clarify that the UNet residual connections are constructed in a way that the decoder layers are connected to the corresponding encoder layers at the same downsampling levels. For example, the decoder layer at 2x downsampling ratio will only receive latent feature map in the corresponding encoder layer that is also downsampled 2x.
>
> > fig6 is hard to understand. for me it is again an example of something showing some information, but for me it is hard to connect it back to the context of video diffusion. in contrast, fig7 is quite instructive.
>
> We agree that some elements in Figure 6 seems redundant (e.g. the 3D spatiotemporal attention part as such attention mechanisms are barely used due to high computational complexity). We remade Figure 6 to focus more on the extension of 3D convolution layers in video diffusion models.
>
> > it is very important to understand how the U-Net/ViT building blocks are actually used in the diffusion model (denoising process i guess). this is briefly described here and there in sec4, however, it would help a lot if the actual connection to the mathematical notation would be made to see how precisely the ViT is used. is it simply the 1-step conditional of the denoising process? or the error (e.g., relating it to the epsilon_theta(x, t) from sec3)? stated differently, one gets an idea about U-net, but what i'm missing is the precise link back to the original problem of vision diffusion. what is a "denoiser"?
>
> Thanks for the suggestions. We added a paragraph in Section 5 to describe how the UNet/ViT models are used in the denoising process for diffusion-based video generation.
>
> [1] DDPM: Ho, J., Jain, A. and Abbeel, P., 2020. Denoising diffusion probabilistic models. Advances in neural information processing systems, 33, pp.6840-6851.

---

> ### Author Response · Authors · 2024-09-18
> **Improvements (part 2)**
>
> > in sec 11.1 i found it very interesting to read that one way to use trained diffusion models for downstream tasks is to use their latent representations. however, it remains unclear to me how this is done, and there should be more of an explanation: given a diffusion model, i.e., denoising models as presented in sec3, how specifically, from this basis, does one obtain the representations? what i don't understand is that the model maps from noise to real, right? however, a representation should map from real to noise, right? there is fig12, but for me it is again an example of showing some aspects, but leaving out the curcial aspects for the reader to make sense out of how this is specifically done for diffusion models.
>
> Thank you for sharing our interest in using diffusion models for various downstream tasks! We are happy to provide further clarification here. The reviewer is right that the denoiser model maps noised inputs to real inputs. A representation should map from real inputs to a low-dimensional vector. This is achieved using a diffusion model by first partially noising real inputs and then taking intermediate outputs from the model (e.g. from the middle of a UNet).
>
> We thank the reviewer again for raising this, and we have further clarified Section 11.1.

---

> > ### Comment · Reviewer_GaEp · 2024-10-07
> >
> > > We condition on by introducing the term in the original Eq. (7) using Bayes' rule:
> > . This ensures that each KL divergence term in Eq. (7) has analytic training objectives for the network at different time steps. This is a commonly-used reparameterization technique to enable efficient training for diffusion models and is proposed in DDPM [1].
> >
> > this is interesting and seems to explain the gap. however, as far as i see in the revised version, the gap is still unexplained. i would strongly recommend to make that section 3.1 as gap-less as possible, since it is the basis for understanding all the rest. so i'd recommend to explicitly add the formal derivation that you indicated now to me (via bayes rule). if you feel that then the section gets too long, consider putting such more detailed derivations into the appendix.
> >
> > generally, i think this section should have clean derivation without any gaps. if some derivations would be overly long, then at least explain them high level, and give a reference to where the details are.
> >
> > otherwise, if you keep such gaps, the section becomes a bit pointless to me.

---

> > > ### Author Response · Authors · 2024-10-09
> > >
> > > Dear Reviewer GaEp,
> > >
> > > Thank you for taking the time to review our rebuttal and provide constructive feedback. We will revise Section 3.1 of the main text to clarify the details and ensure that it is mathematically self-contained.

---

### Review · Reviewer_sifm · 2024-07-11

**Summary Of Contributions:**

The paper presents a survey of recent advances in diffusion models in the video domain. It covers, e.g., model architectures, different application areas, and how to handle temporal dynamics in video streams. The survey is very up-to-date in the sense that a majority of the covered papers are from 2022-2024, which is important since the field is moving forward fast.

The survey provides a good overview over recent advances at a level suitable for a person new to the field, i.e., a reasonable taxonomy and a good division of topics that are covered. Each paper's contributions are summarized briefly but still highlights the main importance of each paper.

**Audience:**

Yes

**Broader Impact Concerns:**

No ethical concerns

**Claims And Evidence:**

Yes

**Requested Changes:**

* Background section of diffusion models should be expended and improved.
* A method section should be included, describing how the survey was conducted etc.
* Make structural changes so figures and tables come after where they are reference from the main text (and make sure all of them are connected to the main text).
* Write a proper summary / discussion section, where trends, insights, general observations, etc. are presented and discussed.

**Strengths And Weaknesses:**

**Strengths**
+ Generally well-written and easy to read.
+ A timely survey that provides a good overview of the area, especially for someone who would like get into the latest advancements
+ Up-to-date reference list that covers the most recent years (2022-2024).
+ Generally well-structured, but with som necessary changes (see below)

**Weaknesses**
- Very shallow background on how diffusion models actually work. The authors claim that they have added Section 3 with more background. However, Section 3 is just a reformatted version of Section 3.1 from the previous submission, no new information as far as I can see.
- A method section is missing, i.e., how was the survey conducted, how were the papers selected, which search terms were used for database searches, which reference databases were searched, etc.
- Several of the Tables and Figures are not referenced from the main text or very far from the figure, e.g., Fig 4 is on page 6 and referenced from page 19, Figs 9, 10, 11 and Table 3 are not referenced at all. In addition, it is quite annoying that the Figures often come before the main text where they are referenced.
- The paper lacks a more general and reflective summary of all the papers discussed. For example, what shall we have learnt from reading the paper, which are the most important trends, methods, etc.
- I was excited in the beginning when reading the paper, but after a while the paper looked more and more as a very large related work section. What I mean is that for each topic, e.g., Section 8 on Video completion, the referenced papers are just summarized in a paragraph (looks like a compressed abstract of each paper) but there is no high-level observation for each topic.
- Unclear what Sections 7.2. and 7.3 contribute (not clear how the techniques in the cited papers work).
- Very unclear what the differences between a data set and benchmark are. In both sections, the same data set a referenced in several cases, e.g, UCF-101, Sky Time-Lapse, MSR-VTT, and Tai-Chi-HD.
- Table 4 would fit much better after the models presented in the Table are described in Sections 7, 8, 9, and 10.
- Section 11 (Video Diffusion Models for Intelligent Decision Making) feels a bit of topic for me, but that might be a personal preference.
- Section 13 (Outlook and Challenges) are short and to some extent a disappointment. No real outlook or reflection is presented, no overall summary of the area or general insights, and only three challenges are listed. Table 3 could be a good starting point for such a section.

---

> ### Author Response · Authors · 2024-09-18
> **Improvements**
>
> Dear reviewer,
>
> thank you very much for your time and thoughtful feedback.
>
> > Very shallow background on how diffusion models actually work. The authors claim that they have added Section 3 with more background. However, Section 3 is just a reformatted version of Section 3.1 from the previous submission, no new information as far as I can see.
>
> We have added more background knowledge in Section 3, especially a new subsection Score-based Model Formulation, to connect the DDPM and score-based models such as Score SDEs. We also improve the writing of the DDPM background to improve clarify.
>
> > A method section is missing, i.e., how was the survey conducted, how were the papers selected, which search terms were used for database searches, which reference databases were searched, etc.
>
> We have added a paragraph in Section 2, Taxonomy of Applications, to explain the methodology we adopted for conducting the literature review.
>
> > Several of the Tables and Figures are not referenced from the main text or very far from the figure, e.g., Fig 4 is on page 6 and referenced from page 19, Figs 9, 10, 11 and Table 3 are not referenced at all. In addition, it is quite annoying that the Figures often come before the main text where they are referenced.
>
> We have relocated and referenced all Tables and Figures in the text.
>
> > The paper lacks a more general and reflective summary of all the papers discussed. For example, what shall we have learnt from reading the paper, which are the most important trends, methods, etc.
>
> We've added more technical insights in the text and introduced the flow-matching method for the next wave of generative modeling in the outlook section.
>
> > I was excited in the beginning when reading the paper, but after a while the paper looked more and more as a very large related work section. What I mean is that for each topic, e.g., Section 8 on Video completion, the referenced papers are just summarized in a paragraph (looks like a compressed abstract of each paper) but there is no high-level observation for each topic.
>
> Thank you for pointing this out. We intended to use the first paragraph of each section to provide some high-level overview and motivation of the topic related to that section. For example, in Section 8, we mentioned in the first paragraph that video diffusion models are limited to generating videos with fixed lengths, and thus motivates the literatures in the section to propose different methods to generate longer video sequences. We have expanded the first paragraph of Section 8 to make the high-level motivation clearer.
>
> > Unclear what Sections 7.2. and 7.3 contribute (not clear how the techniques in the cited papers work).
>
> Thank you for raising this concern. We agree that while literatures in Section 7.2 and 7.3 focus on different aspects of video generation such as training-free methods and personalized generation methods, in general they still belong to the broader categories of text-to-video generation. We have merged Section 7.2 and 7.3 to 7.1.3 in the revision and also added more technical details regarding these methods.
>
> > Very unclear what the differences between a data set and benchmark are. In both sections, the same data set a referenced in several cases, e.g, UCF-101, Sky Time-Lapse, MSR-VTT, and Tai-Chi-HD.
>
> We acknowledge that the term "benchmark" can have different meanings, which may lead to a misunderstanding. In our case, we use it to refer to a data benchmark, i.e. a dataset that is commonly used for evalution of and comparison across models: "Data benchmarks focus on the datasets used in AI training and evaluation. They provide standardized datasets the community can use to train and test models, ensuring a level playing field for comparisons" (https://harvard-edge.github.io/cs249r_book/contents/benchmarking/benchmarking.html#data-benchmarks). This explains why the same datasets appear in the training data and benchmarks sections. We have added a paragraph to the benchmarks section that hopefully clarifies this.
>
> > Table 4 would fit much better after the models presented in the Table are described in Sections 7, 8, 9, and 10.
>
> We relocated Table 4 below the presentation of the models in the text.
>
> > Section 11 (Video Diffusion Models for Intelligent Decision Making) feels a bit of topic for me, but that might be a personal preference.
>
> Thank you for raising this point. We believe that this section is very relevant as it simply another application of video diffusion models for various tasks, and also serves to relate VDMs to the wider machine learning literature. Indeed, Reviewer GaEp noted that they “found it very interesting to read that one way to use trained diffusion models for downstream tasks is to use their latent representations.”

---

### Review · Reviewer_NarQ · 2024-09-04

**Summary Of Contributions:**

This paper provides a very thorough and comprehensive survey of video diffusion models, covering perspectives such as benchmarks, metrics, different conditions for videos, and so on.

**Audience:**

Yes

**Broader Impact Concerns:**

none.

**Claims And Evidence:**

Yes

**Requested Changes:**

see above: weaknesses.

**Strengths And Weaknesses:**

Strengths:
As in the contributions, this paper provides a very thorough survey of video diffusion models.

Weaknesses:
However, the structure of some sections seem a little bit confusing. Particularly in Section 7:
1. Section 7 is focusing on video generation. However, it uses two subsections to introduce GAN and transformer, which is very confusing to me as this is a paper on diffusion models. Maybe the authors could move these to the overview of this section, talking about previous works, instead of specific subsections?
2. I like the authors list out different conditionings in the intro part (e.g., no conditioning, text-conditioned, image-conditioned, audio-conditioned). However, the categories are really messy in the main paper. With 7.1 about both no conditioning and text-conditioned, and GAN and transformer inside 7.1, diffusion models is 7.1.3 (which is really confusing as I'm thinking that it's a paper about diffusion models instead of a subsubsection). Then, 7.4 is about image-conditioned generation, but the whole section 9 is about audio-conditioned generation. My suggestion is that briefly cover GAN and transformer in the overview of Section 7, and different conditionings should be different subsections (e.g., 7.1 no-conditioning, 7.2 text-conditioned, 7.3 image-conditioned etc...) in section 7.


minor: in the benchmarks section, maybe also include human activities dataset (e.g., ego-exo 4d) and robotics datasets (e.g., open-x embodiment)?

---

> ### Author Response · Authors · 2024-09-18
> **Improvements**
>
> Dear reviewer,
>
> thank you very much for your time and thoughtful feedback.
>
> > Weaknesses: However, the structure of some sections seem a little bit confusing. Particularly in Section 7:
> >
> > Section 7 is focusing on video generation. However, it uses two subsections to introduce GAN and transformer, which is very confusing to me as this is a paper on diffusion models. Maybe the authors could move these to the overview of this section, talking about previous works, instead of specific subsections?
> >
> > I like the authors list out different conditionings in the intro part (e.g., no conditioning, text-conditioned, image-conditioned, audio-conditioned). However, the categories are really messy in the main paper. With 7.1 about both no conditioning and text-conditioned, and GAN and transformer inside 7.1, diffusion models is 7.1.3 (which is really confusing as I'm thinking that it's a paper about diffusion models instead of a subsubsection). Then, 7.4 is about image-conditioned generation, but the whole section 9 is about audio-conditioned generation. My suggestion is that briefly cover GAN and transformer in the overview of Section 7, and different conditionings should be different subsections (e.g., 7.1 no-conditioning, 7.2 text-conditioned, 7.3 image-conditioned etc...) in section 7.
>
> We have improved section 7 in line with your suggestions.
>
> >minor: in the benchmarks section, maybe also include human activities dataset (e.g., ego-exo 4d) and robotics datasets (e.g., open-x embodiment)?
>
> Thank you for your suggestion, we have now added these two datasets into the paper!

---

> > ### Comment · Reviewer_NarQ · 2024-10-16
> >
> > Thank you for the revisions. The paper looks good to me now.

---

> > > ### Author Response · Authors · 2024-10-22
> > >
> > > Thank you very much for your review and discussion!

---

> ### Comment · Action_Editor_RY48 · 2024-10-30
> **recommendation**
>
> Dear Reviewer,
>
> Could you make recommendation on this paper?
>
> Thanks!
>
> AE.

---

### Author Response · Authors · 2024-10-16

Dear Reviewers and Action Editor,

We sincerely thank you for your time and efforts in reviewing our manuscript and for providing constructive feedback to improve the quality of our paper. It has now been about four weeks since we submitted our responses to the reviewers' comments.

As the October 2 deadline for the formal decision recommendation has passed, we remain eager to address any further concerns and make additional improvements to the paper as needed. Please let us know if there is anything else we can do to enhance the manuscript or if any points require further clarification.

Thank you once again for your continued support and guidance.

Best regards, \
Authors

---

### Decision · Action_Editor_RY48 · 2024-11-04

**Recommendation:** Accept with minor revision

**Comment:**

I recommend accepting this survey paper on Video Diffusion Models. All three reviewers supported acceptance, with Reviewer GaEp giving a "Leaning Accept," Reviewer sifm recommending "Accept," and Reviewer NarQ indicating satisfaction with revisions. The paper demonstrates strong technical merit through its comprehensive coverage (2022-2024), clear taxonomy, and mathematical foundations. The authors have been highly responsive to feedback, improving organization, adding methodology sections, and enhancing technical clarity. While some concerns remain about mathematical derivation clarity in Section 3.1, these can be addressed in final revision. The paper successfully fills an important gap in the literature and provides valuable consolidation of recent developments, making it valuable for both newcomers and experienced researchers in TMLR's audience.

**Audience:**

TMLR's audience would be interested in this paper. As a comprehensive survey of video diffusion models, it covers core concepts including model architectures, temporal dynamics, evaluation metrics, and various applications. Reviewer sifm noted it as "a timely survey that provides a good overview of the area, especially for someone who would like to get into the latest advancements." The paper consolidates recent developments (2022-2024) in this fast-moving field, making it valuable for both newcomers and experienced researchers.

**Claims And Evidence:**

The paper provides a comprehensive survey of video diffusion models (2022-2024) with mathematical formulations, visual examples, benchmarks, and quantitative analysis. The authors have improved technical clarity through revisions. All three reviewers confirmed the validity of the claims. The consensus indicates the paper's claims are well-supported and technically sound.

---

> ### Author Response · Authors · 2024-11-15
>
> Dear AE and Reviewers,
>
> Thank you so much for taking the time to provide valuable feedback on our paper. We are truly grateful for your constructive comments, which have helped us improve the paper in many ways. As suggested by reviewer GaEp, we have included the mathematical derivations in Section 3.1 for coherence. Please find the latest PDF attached for your reference.
>
> Thank you.